# Agentic Proposing: Enhancing Large Language Model Reasoning via Compositional Skill Synthesis

**Zhengbo Jiao** [1]  **Shaobo Wang** [2]  **Zifan Zhang** [2]  **Xuan Ren** [1]  **Wei Wang** [1]  **Bing Zhao** [1]  **Hu Wei** [1]  **Linfeng Zhang** [2]

## Abstract

Advancing complex reasoning in large language models relies on high-quality, verifiable datasets, yet human annotation remains cost-prohibitive and difficult to scale. Current synthesis paradigms often face a recurring trade-off: maintaining structural validity typically restricts problem complexity, while relaxing constraints to increase difficulty frequently leads to inconsistent or unsolvable instances. To address this, we propose **Agentic Proposing**, a framework that models problem synthesis as a goal-driven sequential decision process where a specialized agent dynamically selects and composes modular reasoning skills. Through an iterative workflow of internal reflection and tool-use, we develop the **Agentic-Proposer-4B** using Multi-Granularity Policy Optimization (MGPO) to generate high-precision, verifiable training trajectories across mathematics, coding, and science. Empirical results demonstrate that downstream solvers trained on agent-synthesized data significantly outperform leading baselines and exhibit robust cross-domain generalization. Notably, a 30B solver trained on only 11,000 synthesized trajectories achieves a state-of-the-art 91.6% accuracy on AIME25, rivaling frontier-scale proprietary models such as GPT-5 and proving that a small volume of high-quality synthetic signals can effectively substitute for massive human-curated datasets.

## 1. Introduction

Advancing complex reasoning, particularly in high-difficulty domains such as mathematical problem-solving,

[1]Alibaba Group [2]Shanghai Jiao Tong University, Shanghai, China. Correspondence to: Zhengbo Jiao <jiaozhengbo.jzb@alibaba-inc.com>, Linfeng Zhang <zhanglinfeng@sjtu.edu.cn>.

*Proceedings of the 43rd International Conference on Machine Learning*, Seoul, South Korea. PMLR 306, 2026. Copyright 2026 by the author(s).

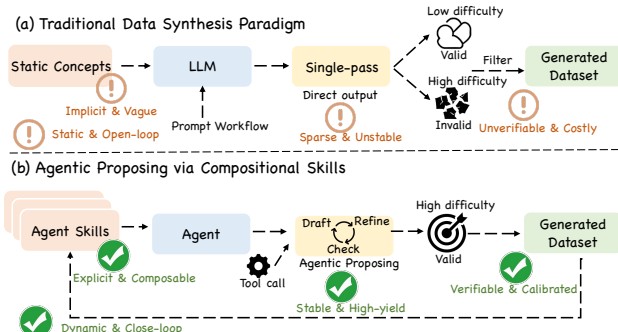

*Figure 1.* Comparison of data synthesis paradigms. (a) Traditional: open-loop, single-pass generation from static concepts—often unstable and unverifiable. (b) Agentic proposing: a closed-loop process that composes modular skills and uses tool-assisted verification to produce stable, verifiable, and well-calibrated problems.

represents a central frontier in large language model (LLM) research. The introduction of OpenAI o1 (OpenAI, 2024) signals a community-wide prioritization of reasoning capabilities as a critical benchmark. Concurrently, breakthroughs like DeepSeek-Math (Shao et al., 2024) have demonstrated the powerful efficacy of reinforcement learning (RL) in unlocking mathematical reasoning potential. However, these RL methods fundamentally depend on verifiable environment feedback, which translates to a critical need for large quantities of high-quality, high-difficulty, and verifiable problems. Currently, sourcing such problems relies heavily on costly and human annotation (Yu et al., 2024). This fundamental limitation has motivated research into methods for synthesizing verifiable, high-difficulty training data.

Current data synthesis paradigms have made substantial strides in scaling and diversifying reasoning datasets. Techniques can be broadly categorized by their core mechanism: some, like *MetaMath* (Yu et al., 2024) and *WizardMath* (Luo et al., 2023), rewrite or evolve existing problems to elicit varied solution pathways; others, such as *MathSmith* (Zhan et al., 2025), *ScaleQuest* (Ding et al., 2024), and key-point-driven methods (Huang et al., 2025b), generate novel questions by extracting and recombining concepts from structured sources; while approaches like *DESIGNER* (Liu et al., 2025b) abstract reusable "design logic" or reasoning patterns from expert problem banks for systematic construction. These methods have expanded dataset scale and can produce

usable instances under their respective settings. However, existing methods typically rely on human-designed generation templates or fixed structural priors to ensure problem validity. This confines problem construction to a narrow, predefined space and hinders the autonomous exploration of novel, high-difficulty reasoning compositions. Conversely, relaxing such constraints to increase generation flexibility often leads to logically inconsistent or unsolvable instances.

To address this challenge, we argue that the synthesis of high-difficulty problems should be viewed not as a monolithic text generation task, but rather as a process of compositional logic engineering. The core motivation is to transform reasoning patterns into executable components that can be orchestrated to explore the reasoning frontier. We introduce the concept of Composable Agent Skills, where problem-construction logic is decomposed into atomic, reasoning modules. Specifically, we propose Agentic Proposing, a framework that models synthesis as a goal-driven sequential decision process. Through an iterative workflow of internal reflection and tool-use, a specialized proposing agent learns to dynamically select and compose these atomic skills to synthesize complex problems that are both logically sound and precisely calibrated in difficulty.

Empirically, the effectiveness of our approach is validated through experiments demonstrating that agent-synthesized trajectories provide superior training signals compared to existing baselines. Notably, a 4B solver trained on only 10,000 synthetic trajectories consistently outperforms established reasoning collections across mathematics, science, and coding. This generalization suggests that high-precision signals, rather than model scale, are the primary bottleneck for reasoning performance. Furthermore, by scaling to a 30B solver, our framework achieves a state-of-the-art 91.6% on AIME 2025, proving that agentic synthesis effectively bridges the gap between open-source models and frontier-scale LLMs. Our principal contributions are as follows:

1. **Agentic Proposing Framework.** We introduce Agentic Proposing, where a specialized agent manages problem synthesis via iterative internal reflection and tool-use, autonomously auditing and correcting errors to improve validity and solvability.
2. **Skill-Based Problem Construction.** We implement a modular pipeline that enables the agent to compose atomic reasoning skills into complex, verifiable problems. This approach supports precise control over logical structure and difficulty.
3. **Agentic Post-training for Proposers.** We train a specialized Agentic-Proposer-4B using agentic SFT and Multi-Granularity Policy Optimization (MGPO), a tailored RL method for data-proposing agents, producing high-fidelity reasoning trajectories.
4. **Superior Empirical Performance.** We demonstrate the

effectiveness of the data synthesized by our proposer. By training on agent-generated problems, downstream 4B solvers outperform leading baselines and show strong performance in science and coding. Notably, a 30B solver trained on only 11,000 trajectories achieves 91.6% on AIME25, surpassing Grok-4.1-Fast and Claude-4.5-Opus while rivaling GPT-5 and Gemini-3-Pro.

## 2. Related Work

**Seed-Based Expansion.** A line of work focuses on augmenting training data by iteratively expanding a small set of seed questions. Early methods such as Self-Instruct (Wang et al., 2023) generate new instructions from model outputs. Subsequently, WizardMath (Luo et al., 2023) and Auto Evol-Instruct (Zeng et al., 2024) evolve instructions to increase complexity and diversity. More recent approaches include MetaMath (Yu et al., 2024), which bootstraps mathematical questions, and CoT-Self-Instruct (Yu et al., 2025) PromptCoT (Zhao et al., 2025b),PromptCoT 2.0 (Zhao et al., 2025b) which incorporates chain-of-thought reasoning. While these methods effectively expand data, they remain constrained by seed quality and lack mechanisms to dynamically adapt the generated curriculum to a model's evolving capabilities.

**Corpus-Based Extraction.** Another approach synthesizes questions directly from text corpora or structured knowledge sources. ScaleQuest (Ding et al., 2024) proposes a scalable framework for question synthesis. MathSmith (Zhan et al., 2025) and Key-point-driven Data Synthesis (Huang et al., 2025b) generate challenging mathematical problems through reinforced policies. DESIGNER (Liu et al., 2025b) employs design-logic-guided synthesis for multidisciplinary reasoning data. While this approach ensures broad factual grounding, it often struggles to precisely calibrate question difficulty and may fail to target specific learner weaknesses.

**Agent-Based Self-Play.** Recent work explores using language models in self-play setups to generate reasoning data. DeepSeekMath (Shao et al., 2024) and DeepSeek-R1 (Guo et al., 2025) employ reinforcement learning with self-play. R-Zero (Huang et al., 2025a), Absolute Zero (Zhao et al., 2025a), and SPICE (Liu et al., 2025a) further investigate self-evolving reasoning from minimal data. Socratic-Zero (Wang et al., 2025) introduces a co-evolutionary framework with Teacher, Solver, and Generator agents. DataEnvGym (Khan et al., 2025) frames data generation as a sequential decision-making task with student feedback. While these approaches advance automated data generation, they typically rely on static prompting strategies or single-agent self-play architectures, without the explicit POMDP formulation, dynamic skill management ($\tau^{\text{edit}}$), and structured internal reflection/refinement loop (Draft-Check-Refine) that characterize our framework's reactive adaptation capabili-

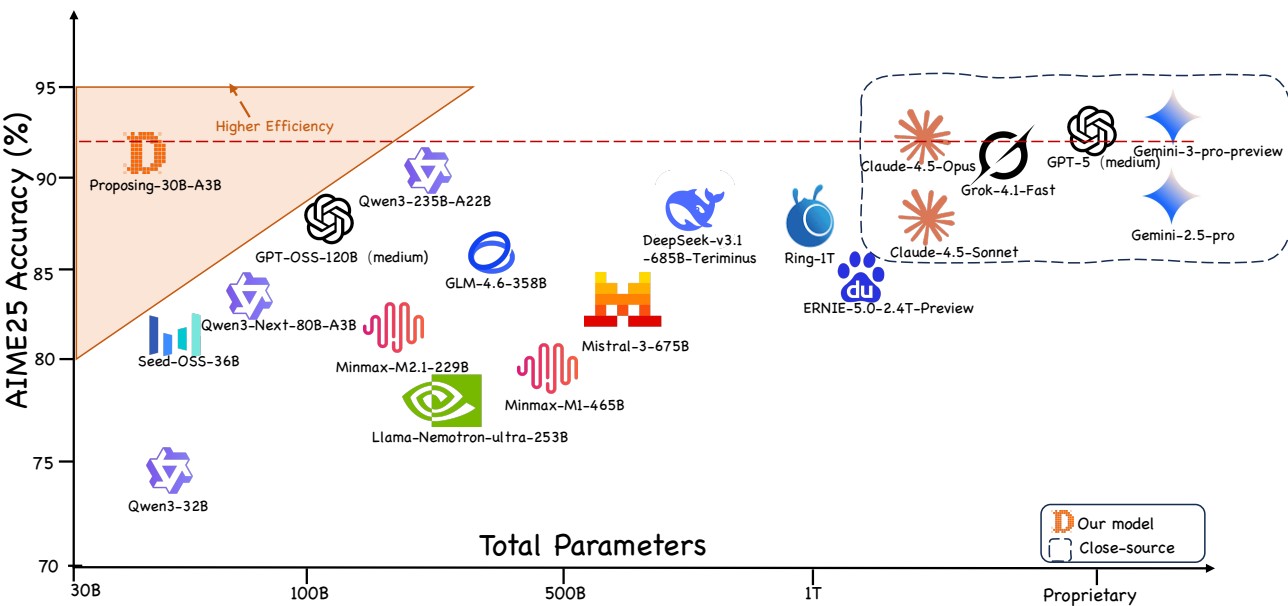

*Figure 2.* Comparison of model scale versus AIME 2025 accuracy (mean@64). Our Proposing-30B-A3B achieves a state-of-the-art 91.6% accuracy, outperforming open-source models with up to 20× more parameters (e.g., DeepSeek-v3.1, Mistral-3) and rivaling top-tier proprietary models. This demonstrates the superior parameter efficiency unlocked by our agentic synthesis framework.

ties.

## 3. The Agentic Proposing Framework

### 3.1. Problem Formulation and Skill Composition

We conceptualize problem synthesis as an iterative search for logical consistency and complexity in a high-dimensional reasoning space. This task is modeled as a **Partially Observable Markov Decision Process (POMDP)**, defined by the tuple $(\mathcal{S}, \mathcal{A}, \mathcal{O}, P, R, \gamma)$. In this framework, $\mathcal{S}$ is the latent state space representing the underlying logical integrity and difficulty of a problem; $\mathcal{A}$ is the action space; $\mathcal{O}$ is the observation space; $P : \mathcal{S} \times \mathcal{A} \to \Delta(\mathcal{S})$ denotes the transition dynamics, where $\Delta(\mathcal{S})$ is the probability simplex over $\mathcal{S}$; $R : \mathcal{S} \times \mathcal{A} \to \mathbb{R}$ is the reward function; and $\gamma \in [0, 1)$ is the discount factor. Our choice of a POMDP formulation is grounded in a critical insight: **the logical solvability of a synthesized problem is a latent property $\mathcal{S}$ that remains intrinsically unobservable through surface dialogue history alone.** Consequently, the agent must actively "probe" the environment—using tool-use and internal reflection—to reduce uncertainty and converge on a valid problem instance.

**Observation and Context.** To equip the agent with diverse construction patterns, we first initialize an autonomous skill library $\mathcal{K}_{\text{self}}$ consisting of atomic reasoning modules (see Section 2.3). At each time step $t$, the agent receives an observation $o_t \in \mathcal{O}$ defined as:

$$o_t = \langle \mathcal{K}_t, h_t, \sigma_t \rangle, \quad (1)$$

where $\mathcal{K}_t \subseteq \mathcal{K}_{\text{self}}$ is the currently active subset of skills, and $h_t$ is the dialogue history including prior tool outputs. Crucially, we introduce $\sigma_t$ as a **stage indicator** that tracks the agent's progress through semantic phases such as *drafting*, *checking*, and *refining*. Rather than being constrained by a state machine, the agent utilizes $\sigma_t$ as a functional context to adaptively navigate the synthesis process—for instance, proactively returning to a refinement mode if a logical flaw is uncovered during validation—thereby maintaining a self-corrective loop.

**Action Space.** The action space is partitioned into three functional domains, $\mathcal{A} = \mathcal{U} \cup \mathcal{T} \cup \mathcal{F}$, where:

1. **Internal Actions $\mathcal{U}$:** The set of natural language responses, including an **internal reflection action** $\tau^{\text{think}}$ used to generate reasoning chains for logical auditing before committing to an observable output.

2. **Interactive Tools $\mathcal{T}$:** The set of tool invocations, including $\tau^{\text{exec}}$ for sandboxed code execution and $\tau^{\text{edit}}$ for dynamic skill pruning. Specifically, $\tau^{\text{edit}}$ allows the agent to update the active set $\mathcal{K}_t$ by executing $\Delta \mathcal{K} = \{-k\}$ to autonomously remove a misaligned skill $k$.

3. **Terminal Submission $\mathcal{F}$:** The action $(\tau^{\text{submit}}, q)$ used to submit the final synthesized problem $q$ within the problem space $\mathcal{Q}$.

**Skill Composition.** Grounded in the principle of emergent compositionality (Yuan et al., 2025), we introduce the following lemma to justify our modular design:

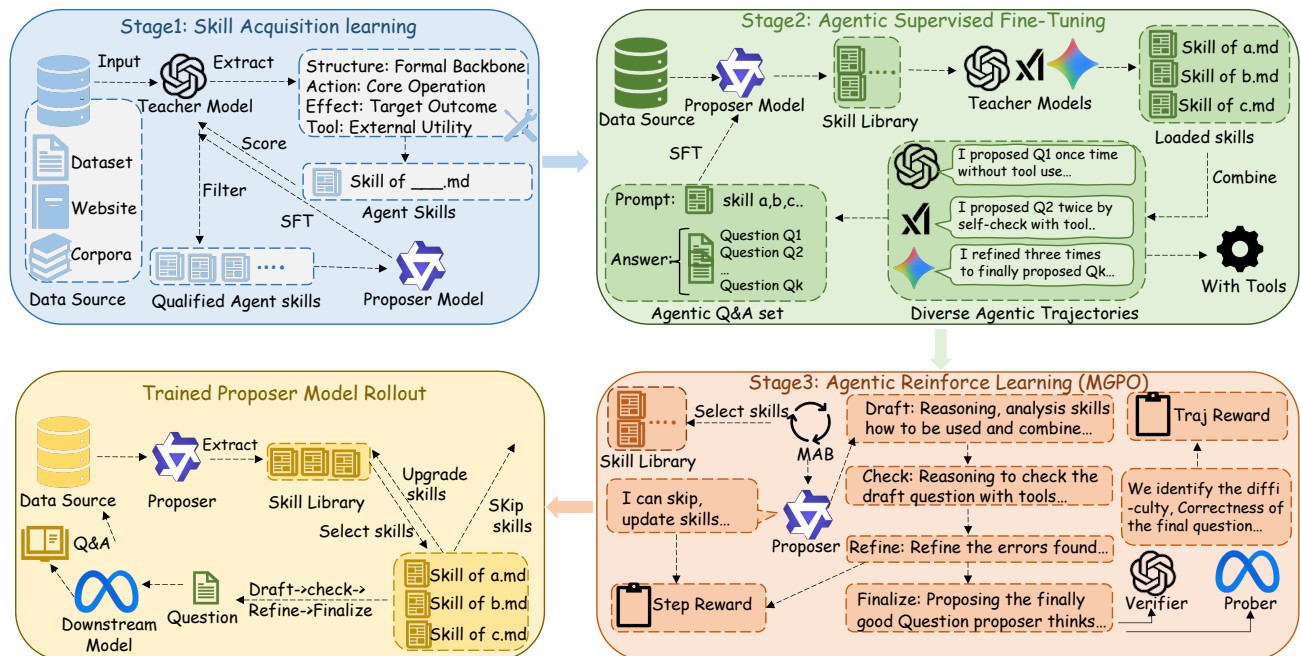

*Figure 3.* **The Agentic-Proposer Synthesis Pipeline.** The framework evolves through three sequential phases: **(Stage 1) Skill Acquisition**: extracting and filtering atomic skills from diverse corpora to build an autonomous skill library. **(Stage 2) Agentic SFT**: mimicking expert trajectories that incorporate internal reflection, tool execution, and dynamic skill pruning. **(Stage 3) Agentic RL (MGPO)**: optimizing the policy via multi-granularity rewards. The agent follows a structured reasoning flow (Draft → Check → Refine → Finalize), guided by a curriculum-based skill distribution to synthesize high-difficulty, logically sound problems for downstream solver training.

**Lemma 3.1** (Emergent Skill Compositionality). *Let $\mathcal{F}$ be a set of atomic skills. For a compositional task $h = g \circ f$, if a reinforcement learning objective provides positive rewards only when the output matches $h(x)$, an agent can learn to orchestrate $g$ and $f$ to solve $h$ with high probability, even if the specific composition was unseen during pre-training.*

Based on this, we formalize each skill $k \in \mathcal{K}_{\text{self}}$ as a structured attribute tuple $k \triangleq \langle \iota, \mu, \delta, \tau \rangle$, encoding its reasoning intent $\iota$, construction method $\mu$, difficulty effect $\delta$, and tool-use hint $\tau$. We define a mapping operator $\Phi : \mathcal{K}^n \to \mathcal{L}$ that transforms a composition of $n$ selected skills into natural language constraints in a high-dimensional instruction space $\mathcal{L}$. The problem generation is then governed by the policy $\pi_\theta$, parameterized by $\theta$:

$$q \sim \pi_\theta\big(\cdot \mid o_t, \Phi(k_1, \ldots, k_n), \mathcal{K}_t\big). \quad (2)$$

### 3.2. Overall Pipeline

Our training pipeline, illustrated in **Figure 3**, orchestrates the evolution of the Proposer through three phases:

1. **Skill Acquisition and Library Formalization**: We distill and formalize a diverse set of atomic skills from large-scale corpora to construct the foundational skill library $\mathcal{K}_{\text{self}}$ that serves as the agent's prior knowledge.

2. **Agentic Supervised Fine-tuning (SFT)**: We leverage a teacher policy to synthesize expert trajectories exhibiting complex behaviors—such as internal reflection and tool-use—to initialize the agent's policy $\pi_\theta$ via behavioral cloning.

3. **Agentic Post-training via MGPO**: To bridge the gap between logical validity and extreme difficulty, we employ the Multi-Granularity Policy Optimization (MGPO) algorithm to refine the agent's ability to orchestrate modular skills into high-precision, verifiable, and challenging tasks.

### 3.3. Skill Acquisition and Dynamic Pruning

As depicted in **Stage 1 of Figure 3**, let $\mathcal{D}_{\text{corpus}}$ be a mixed-source corpus. A teacher policy $\pi_{\text{teacher}}$ induces a candidate skill set $\mathcal{K}_{\text{cand}}$. The teacher assigns a quality score $r(k) \in \mathbb{R}$ to each skill $k \in \mathcal{K}_{\text{cand}}$. Through rejection sampling with a threshold $\tau_r \in \mathbb{R}$, we define a filtered skill distribution:

$$p_{\text{skill}}(k) = \frac{\mathbb{I}[r(k) \geq \tau_r] \cdot \pi_{\text{teacher}}(k \mid \mathcal{D}_{\text{corpus}})}{\sum_{k' \in \mathcal{K}_{\text{cand}}} \mathbb{I}[r(k') \geq \tau_r] \cdot \pi_{\text{teacher}}(k' \mid \mathcal{D}_{\text{corpus}})}, \quad (3)$$

where $\mathbb{I}[\cdot]$ denotes the indicator function. A proposer agent is trained via a maximum likelihood objective:

$$\mathcal{L}_{\text{skill-acq}}(\theta) = -\mathbb{E}_{k \sim p_{\text{skill}}} \left[ \log \pi_\theta(k \mid \mathcal{D}_{\text{corpus}}) \right], \quad (4)$$

thereby building the autonomous skill library $\mathcal{K}_{\text{self}}$.

**Dynamic Pruning Mechanism.** At any stage of the synthesis process, particularly during the drafting stage, the agent can invoke the internal reflection action $\tau^{\text{think}}$ to evaluate the suitability of the currently active skill set $\mathcal{K}_t$. If the agent predicts that a skill $k \in \mathcal{K}_t$ is misaligned with the task objective or likely to cause logical errors, it proactively executes the tool call $\tau^{\text{edit}}$ to remove that skill from $\mathcal{K}_t$. This forward-looking self-correction mechanism ensures the robustness of the synthesis process by preventing low-quality generation paths at their source.

## 3.4. Agentic Supervised Fine-tuning

In Stage 2, the goal is to teach the model to imitate an expert's complex decision-making process. We use a teacher policy to generate a high-quality dataset of agentic trajectories, $\mathcal{D}_{\text{expert}}$. Each trajectory $\tau \in \mathcal{D}_{\text{expert}}$ is a sequence of observations and actions $\tau = \{(o_t, a_t)\}_{t=1}^{T_\tau}$ of length $T_\tau$, containing rich agentic behaviors such as internal reflections, tool calls, and skill pruning.

To ensure the quality of the demonstrations, all final problems $q \in \mathcal{Q}$ synthesized within $\mathcal{D}_{\text{expert}}$ are rigorously filtered by a high-precision verifier $\mathcal{V} : \mathcal{Q} \rightarrow \{0, 1\}$, where $\mathcal{V}(q) = 1$ signifies that the problem is logically consistent and admits a correct solution. We define a binary validity indicator $\mathbb{I}_{\text{valid}}(q) = \mathbb{I}[\mathcal{V}(q) = 1]$. Only trajectories whose final problems satisfy $\mathbb{I}_{\text{valid}} = 1$ are retained, forming the final SFT dataset $\mathcal{D}_{\text{SFT}}$.

By performing **behavioral cloning** on this dataset, we obtain the reference policy $\pi_{\text{ref}}$ for the subsequent RL phase, where $\pi_{\text{ref}}$ is the policy $\pi_\theta$ that minimizes the following cross-entropy loss:

$$\mathcal{L}_{\text{SFT}}(\theta) = -\frac{1}{|\mathcal{D}_{\text{SFT}}|} \sum_{\tau \in \mathcal{D}_{\text{SFT}}} \sum_{t=1}^{T_\tau} \log \pi_\theta(a_t \mid o_t). \quad (5)$$

## 3.5. Agentic Reinforcement Learning

The final phase employs the **Multi-Granularity Policy Optimization (MGPO)** algorithm to further optimize the agent's policy, as illustrated in Stage 3 of **Figure 3**.

### 3.5.1. CURRICULUM-BASED SKILL DISTRIBUTION

To dynamically focus training on skill categories where the agent underperforms, we introduce a curriculum learning mechanism. Let $\mathcal{C}$ be the set of skill categories. The system maintains a proficiency vector $\mathbf{m} \in [0, 1]^{|\mathcal{C}|}$ over all categories. After each iteration, the proficiency $m_c$ for category $c \in \mathcal{C}$ is updated via an Exponential Moving Average (EMA):

$$m_c^{(t+1)} = (1 - \alpha)m_c^{(t)} + \alpha \cdot \text{success\_rate}_c^{(t)}, \quad (6)$$

where $\alpha \in (0, 1)$ is a smoothing factor and $\text{success\_rate}_c$ is the verifier-validated pass rate for category $c$. In the subsequent iteration, skill categories are sampled for problem synthesis with a probability inversely proportional to their proficiency: $p(c) \propto 1/(m_c + \epsilon)$, where $\epsilon > 0$ is a small constant to prevent division by zero.

### 3.5.2. LAYERED REWARD FUNCTION

We utilize a layered reward structure corresponding to the *Step* and *Trajectory* feedback loops. Let $V(q) \in \{0, 1\}$ be the binary output of the verifier $\mathcal{V}$, and $\rho(q) \in [0, 1]$ be the success rate (Pass@$k$) estimated by an external prober $\mathcal{P}$. The terminal reward $r_T$ is defined as:

$$r_T = V(q) \cdot (R_{\text{base}} + \mathbb{I}[\rho(q) > 0] \cdot \lambda(1 - \rho(q))), \quad (7)$$

where $R_{\text{base}}$ is the base reward for logical validity and $\lambda > 0$ is a difficulty scaling factor. To provide denser supervision, we supplement this with **intermediate process rewards** $r_t^{\text{proc}} \geq 0$, assigned for successful tool executions ($\tau^{\text{exec}}$) or logically coherent reflections ($\tau^{\text{think}}$). This design ensures that: (i) invalid problems receive zero total reward; (ii) difficulty bonuses are only awarded for solvable instances ($\rho(q) > 0$).

### 3.5.3. MULTI-GRANULARITY POLICY OPTIMIZATION (MGPO)

MGPO addresses the KL-constrained reward maximization problem through a variational reformulation, leading to the following propositions:

**Proposition 3.2** (Optimal Policy Form). *The KL-constrained reward maximization objective:*

$$\max_{\pi_\theta} \mathbb{E}_{o,a \sim \pi_\theta}[R(o, a)] - \beta D_{\text{KL}}[\pi_\theta(\cdot|o)\|\pi_{ref}(\cdot|o)]$$

*has a unique closed-form optimal solution* $\pi^*(a|o) = \frac{1}{Z(o)}\pi_{ref}(a|o)e^{R(o,a)/\beta}$, *where $\beta > 0$ is the KL penalty coefficient and $Z(o)$ is the partition function.*

**Proposition 3.3** (Zero-Sum Weighting Property). *Define the implicit reward as $R_\theta(a|o) = \beta \log(\pi_\theta(a|o)/\pi_{ref}(a|o))$. By constructing sample weights $w(o, a)$ as the difference between centered advantages ($A$) and centered implicit rewards ($\bar{\cdot}$) denotes the group-wise mean):*

$$w(o, a) = (A(o, a) - \bar{A}) - (R_\theta(a|o) - \bar{R}_\theta), \quad (8)$$

*these weights satisfy the **zero-sum property** ($\mathbb{E}[w] = 0$), ensuring that the intractable $Z(o)$ cancels out.*

**Multi-Granularity Advantage Estimation.** MGPO performs group standardization at two granularities to balance global terminal signals with local process feedback:

1. **Trajectory-level Advantage** $A_E(\tau_i)$: terminal reward $(r_T^{(i)} - \bar{r}_T)/\sigma_{r_T}$ within the batch group $\mathcal{G}_{\text{traj}}$.

2. **Stage-level Advantage** $A_S(a_t)$: Standardized process reward $(r_t^{\text{proc}} - \bar{r}^{\text{proc}})/\sigma_{r^{\text{proc}}}$ within subgroups $\mathcal{G}_\sigma$ sharing the same stage indicator $\sigma_t$.

The fused advantage is defined as $A_{i,t}^{\text{fused}} = A_E(\tau_i) + \omega \cdot A_S(a_t)$, where $\omega > 0$ is a fusion weight.

**Final Optimization Objective.** Defining the centered fused advantage $\tilde{A} = A_{i,t}^{\text{fused}} - \text{Mean}(A^{\text{fused}} \mid \mathcal{G}_\sigma)$ and centered implicit reward $\tilde{R}_\theta = R_\theta(a_t) - \text{Mean}(R_\theta \mid \mathcal{G}_\sigma)$, the practical weight $w_{i,t} = \tilde{A} - \tilde{R}_\theta$ is modulated by an asymmetric hyperbolic secant gate for training stability:

$$w'_{i,t} = \text{sech}^2\left(\frac{\tau_{i,t}}{2}(r_{i,t}(\theta) - 1)\right) \cdot w_{i,t}, \qquad (9)$$

where $r_{i,t}(\theta) = \pi_\theta(a_t \mid o_t)/\pi_{\text{old}}(a_t \mid o_t)$ is the importance ratio. The temperature $\tau_{i,t}$ is set asymmetrically: $\tau_{i,t} = \tau_{\text{pos}}$ if $w_{i,t} \geq 0$, and $\tau_{i,t} = \tau_{\text{neg}}$ if $w_{i,t} < 0$, with $\tau_{\text{neg}} > \tau_{\text{pos}}$. The policy is updated via token-normalized weighted maximum likelihood (with total tokens $N$):

$$\mathcal{L}_{\text{MGPO}}(\theta) = -\frac{1}{N}\sum_{i,t,j} w'_{i,t} \log \pi_\theta(x_{i,t,j} \mid o_t^{(i)}, a_{t,<j}^{(i)}). \qquad (10)$$

## 4. Experiments

### 4.1. Experiment Setup

**Models.** Our framework employs a specialized set of models for proposing, verification, and difficulty estimation. Full specifications for the Verifier Ensemble and Prober are provided in Appendix A.2.

**Benchmarks.** We evaluate on contest mathematics (AIME 2024/2025, HMMT, AMO-Bench), algorithmic coding (LiveCodeBench v5/v6), and scientific reasoning (MMLU-Redux/Pro, GPQA, SuperGPQA, OlympicArena). Complete benchmark descriptions are in Appendix A.

**Evaluation Protocols.** Detailed evaluation protocols are specified in Appendix A.1.

**Datasets and Baselines.** All methods are compared under a fixed budget of 10,000–11,000 trajectories against synthetic, human-curated, and frontier-model baselines. Full baseline configurations are in Appendix A.4.

**Training Settings.** Downstream solvers are optimized using GRPO (DeepSeek-AI et al., 2025). Training hyperparameters and protocols are detailed in Appendix A.3.

### 4.2. Main Results

**Performance on Contest Mathematics.** As shown in Figure 4 and Table 1, training a 4B-parameter solver on a fixed budget of 10,000 synthesized mathematics trajectories

yields a significant overall gain of +4.1 points. Notably, while many established baselines such as *MetaMath* and *OpenR1math* exhibit performance degradation relative to the zero-shot baseline, our framework achieves consistent improvements. The most substantial gains are observed in high-difficulty contests: +4.5 points on AIME 2025 and +5.5 points on HMMT.

A closer examination of Table 1 reveals a consistent pattern across baseline categories. *Template-driven* methods (*MetaMath*, *WizardMath*) degrade performance by $-3.2$ to $-3.4$ points, suggesting that rigid algebraic transformations introduce distribution mismatch with the target solver. *Human-curated* datasets show mixed results: *Deepmath* and *Polaris* achieve moderate gains (+1.7 and +1.5 respectively) but remain well below our method, indicating that even expert-authored datasets lack sufficient density of verifiable reasoning chains. Most strikingly, frontier-model-generated problems (*GPT-5.2-High*, *Gemini-3-Pro*, *Claude4.5-Opus*) all underperform the zero-shot baseline by $-1.2$ to $-2.8$ points, despite the enormous capacity of the generators. This underscores a critical insight: model scale alone is insufficient for high-quality problem synthesis—structured compositional skill guidance and iterative verification are indispensable.

**Model Scaling and Algorithmic Generalization.** To explore the scalability of our approach, we evaluate a 30B-parameter solver trained on 11,000 mixed trajectories (math and code). As reported in Table 2, our model establishes a new state-of-the-art for its scale, achieving 91.6% on AIME 2025—an absolute improvement of +6.6 points over the zero-shot baseline. Beyond mathematics, the model demonstrates remarkable generalization to competitive programming, with gains of +5.3 and +5.2 points on LiveCodeBench v5 and v6, respectively. This performance surpasses leading open-source reasoning collections such as *OpenMathReasoning* and *PromptCoT 2.0*.

Importantly, the 30B results reveal that our framework's advantages amplify with model scale. While the overall gain for the 4B solver is +4.1 points, the 30B solver achieves +5.9 points under a comparable trajectory budget. We attribute this to the stronger capacity of larger models to internalize the compositional reasoning structures embedded in our synthesized trajectories.

### 4.3. Cross-Domain Robustness and Transfer

**General and Scientific Reasoning.** To assess the transferability of our synthesis pipeline, we evaluate a 4B solver trained on 10,000 trajectories specifically targeting coding and scientific domains. As summarized in Table 3, the model achieves an overall gain of +5.3 points across multidisciplinary benchmarks. Most notably, we observe significant breakthroughs in graduate-level reasoning: +7.3

*Table 1.* Each method uses a 10,000-trajectory training budget and the same GRPO recipe. Evaluation uses Mean@64 accuracy across benchmarks. Overall represents the unweighted average. Arrows ($\uparrow$ / $\downarrow$) show absolute accuracy change vs. the zero-shot baseline.

| Method | AIME24 | AIME25 | HMMT_Feb | AMO-Bench | Overall |
|---|---|---|---|---|---|
| *Qwen3-4B-Instruct-2507 (Baseline Model)* | | | | | |
| +zero-shot | 49.8 | 46.7 | 31.0 | 9.3 | 34.2 |
| *Data Synthesis Methods* | | | | | |
| +Metamath | $44.6^{\downarrow 5.2}$ | $43.5^{\downarrow 3.2}$ | $27.4^{\downarrow 3.6}$ | $7.8^{\downarrow 1.5}$ | $30.8^{\downarrow 3.4}$ |
| +Wizardmath | $44.8^{\downarrow 5.0}$ | $44.0^{\downarrow 2.7}$ | $27.8^{\downarrow 3.2}$ | $7.6^{\downarrow 1.7}$ | $31.0^{\downarrow 3.2}$ |
| +PromptCoT | $50.1^{\uparrow 0.3}$ | $47.3^{\uparrow 0.6}$ | $33.1^{\uparrow 2.1}$ | $9.1^{\downarrow 0.2}$ | $34.9^{\uparrow 0.7}$ |
| +PromptCot 2.0 | $50.9^{\uparrow 1.1}$ | $48.5^{\uparrow 1.8}$ | $34.2^{\uparrow 3.2}$ | $10.2^{\uparrow 0.9}$ | $36.0^{\uparrow 1.8}$ |
| +NuminaMath | $47.3^{\downarrow 2.5}$ | $43.9^{\downarrow 2.8}$ | $29.1^{\downarrow 1.9}$ | $8.2^{\downarrow 1.1}$ | $32.1^{\downarrow 2.1}$ |
| +MathSmith | $50.3^{\uparrow 0.5}$ | $47.1^{\uparrow 0.4}$ | $32.9^{\uparrow 1.9}$ | $9.1^{\downarrow 0.2}$ | $34.8^{\uparrow 0.6}$ |
| *Human-Annotated Methods* | | | | | |
| +OpenR1math | $49.4^{\downarrow 0.4}$ | $45.8^{\downarrow 0.9}$ | $31.8^{\uparrow 0.8}$ | $8.9^{\downarrow 0.4}$ | $34.0^{\downarrow 0.2}$ |
| +Deepmath | $51.2^{\uparrow 1.4}$ | $48.2^{\uparrow 1.5}$ | $33.7^{\uparrow 2.7}$ | $10.3^{\uparrow 1.0}$ | $35.9^{\uparrow 1.7}$ |
| +Polaris | $51.7^{\uparrow 1.9}$ | $47.4^{\uparrow 0.7}$ | $33.8^{\uparrow 2.8}$ | $9.8^{\uparrow 0.5}$ | $35.7^{\uparrow 1.5}$ |
| +OpenthoughtsS3 | $49.6^{\downarrow 0.2}$ | $46.1^{\downarrow 0.6}$ | $30.6^{\downarrow 0.4}$ | $8.4^{\downarrow 0.9}$ | $33.7^{\downarrow 0.5}$ |
| *Agent-Based Self-Play* | | | | | |
| +R-zero | $45.8^{\downarrow 4.0}$ | $44.3^{\downarrow 2.4}$ | $30.1^{\downarrow 0.9}$ | $6.7^{\downarrow 2.6}$ | $31.7^{\downarrow 2.5}$ |
| +Socratic-zero | $50.2^{\uparrow 0.4}$ | $46.9^{\uparrow 0.2}$ | $31.3^{\uparrow 0.3}$ | $9.1^{\downarrow 0.2}$ | $34.4^{\uparrow 0.2}$ |
| *SOTA Model Generated Problems* | | | | | |
| +GPT-5.2-High | $47.7^{\downarrow 2.1}$ | $45.8^{\downarrow 0.9}$ | $29.9^{\downarrow 1.1}$ | $7.8^{\downarrow 1.5}$ | $32.8^{\downarrow 1.4}$ |
| +Gemini-3-Pro | $45.5^{\downarrow 4.3}$ | $43.1^{\downarrow 3.6}$ | $28.7^{\downarrow 2.3}$ | $8.1^{\downarrow 1.2}$ | $31.4^{\downarrow 2.8}$ |
| +Qwen3-Max | $46.2^{\downarrow 3.6}$ | $43.4^{\downarrow 3.3}$ | $30.2^{\downarrow 0.8}$ | $8.5^{\downarrow 0.8}$ | $32.1^{\downarrow 2.1}$ |
| +Claude4.5-Opus | $48.3^{\downarrow 1.5}$ | $45.7^{\downarrow 1.0}$ | $30.8^{\downarrow 0.2}$ | $7.4^{\downarrow 1.9}$ | $33.0^{\downarrow 1.2}$ |
| +DeepSeek-V3.2-Spe | $49.5^{\downarrow 0.3}$ | $45.6^{\downarrow 1.1}$ | $31.2^{\uparrow 0.2}$ | $8.8^{\downarrow 0.5}$ | $33.8^{\downarrow 0.4}$ |
| *Agentic-Proposer-4B Generated Problems* | | | | | |
| **+Agentic Proposing (Ours)** | $\mathbf{53.6}^{\uparrow 3.8}$ | $\mathbf{51.2}^{\uparrow 4.5}$ | $\mathbf{36.5}^{\uparrow 5.5}$ | $\mathbf{11.8}^{\uparrow 2.5}$ | $\mathbf{38.3}^{\uparrow 4.1}$ |

*Table 2.* All methods are trained on 11,000 mixed trajectories using the GRPO recipe. Math benchmarks are evaluated with Mean@64. LCB (LiveCodeBench) v5 and v6 are evaluated with Best-of-5. Arrows denote absolute accuracy change vs. the zero-shot baseline.

| Method | AIME24 | AIME25 | HMMT | LCB v5 | LCB v6 | Overall |
|---|---|---|---|---|---|---|
| *Qwen3-30B-A3B-Thinking-2507 (Baseline Model)* | | | | | | |
| + zero-shot | 87.7 | 85.0 | 71.4 | 68.1 | 66.0 | 75.6 |
| *Baselines* | | | | | | |
| + OpenCodeReasoning | $85.0^{\downarrow 2.7}$ | $81.1^{\downarrow 3.9}$ | $64.9^{\downarrow 6.5}$ | $70.8^{\uparrow 2.7}$ | $67.4^{\uparrow 1.4}$ | $73.8^{\downarrow 1.8}$ |
| + OpenMathReasoning | $87.9^{\uparrow 0.2}$ | $86.1^{\uparrow 1.1}$ | $72.2^{\uparrow 0.8}$ | $65.7^{\downarrow 2.4}$ | $58.4^{\downarrow 7.6}$ | $74.1^{\downarrow 1.5}$ |
| + PromptCoT 2.0 | $92.1^{\uparrow 4.4}$ | $89.8^{\uparrow 4.8}$ | $76.7^{\uparrow 5.3}$ | $74.2^{\uparrow 6.1}$ | $71.0^{\uparrow 5.0}$ | $80.8^{\uparrow 5.2}$ |
| + OpenThoughts-S3 | $85.7^{\downarrow 2.0}$ | $84.7^{\downarrow 0.3}$ | $70.0^{\downarrow 1.4}$ | $68.3^{\uparrow 0.2}$ | $67.2^{\uparrow 1.2}$ | $75.2^{\downarrow 0.4}$ |
| + OpenR1 | $86.3^{\downarrow 1.4}$ | $84.9^{\downarrow 0.1}$ | $68.9^{\downarrow 2.5}$ | $68.3^{\uparrow 0.2}$ | $63.7^{\downarrow 2.3}$ | $74.4^{\downarrow 1.2}$ |
| *Agentic-Proposer-30B Generated Problems* | | | | | | |
| **+ Agentic Proposing (Ours)** | $\mathbf{93.5}^{\uparrow 5.8}$ | $\mathbf{91.6}^{\uparrow 6.6}$ | $\mathbf{77.6}^{\uparrow 6.2}$ | $\mathbf{73.4}^{\uparrow 5.3}$ | $\mathbf{71.2}^{\uparrow 5.2}$ | $\mathbf{81.5}^{\uparrow 5.9}$ |

*Table 3.* Cross-domain generalization. 4B solver trained on 10,000 domain-specific trajectories. Mean@1 accuracy reported.

| | OlyAr | MMLU-R | MMLU-P | GPQA | SGPQA | Avg. |
|---|---|---|---|---|---|---|
| Zero-shot | 42.8 | 84.1 | 69.6 | 62.0 | 42.8 | 56.1 |
| **Ours** | $\mathbf{47.2}^{\uparrow 4.4}$ | $\mathbf{87.3}^{\uparrow 3.2}$ | $\mathbf{75.2}^{\uparrow 5.6}$ | $\mathbf{68.3}^{\uparrow 6.3}$ | $\mathbf{50.1}^{\uparrow 7.3}$ | $\mathbf{61.4}^{\uparrow 5.3}$ |

points on SuperGPQA and +6.3 points on GPQA. These gains are particularly noteworthy because SuperGPQA and GPQA test expert-level reasoning in physics, chemistry, and biology—domains far removed from the mathematical training signal used in Tables 1–2. The result confirms that our Skill Library's compositional structure captures transferable reasoning primitives (e.g., *proof by contradiction*, *dimensional analysis*, *constraint decomposition*) rather than domain-specific heuristics.

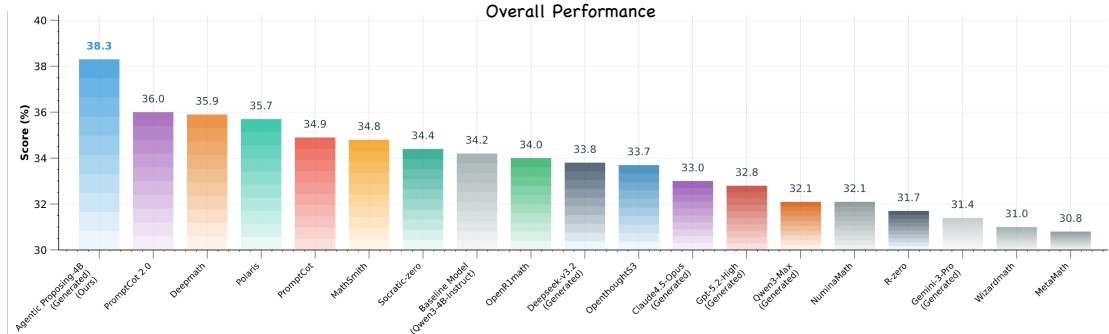

*Figure 4.* Overall performance comparison on contest mathematics benchmarks. Our Agentic Proposing-4B (38.3%) significantly outperforms all baselines under a fixed 10,000-trajectory budget, achieving a +4.1% absolute gain over the zero-shot baseline (34.2%).

*Table 4.* **Generalization to unseen domains.** The 4B Proposer synthesizes data for tasks outside its pre-training distribution.

| Benchmark | Zero-shot | Ours (+10k) | Reference |
|---|---|---|---|
| CL-Bench | 10.7 | **13.9** | Qwen3-Max: 14.2 |
| BrowseComp | <30.0 | **39.7** | DeepResearch: 43.4 |

**OlympicArena.** The performance on the text-only validation subset of OlympicArena (+4.4 points) further confirms that Agentic Proposing enables the acquisition of deep reasoning capabilities. Unlike traditional augmentation that often leads to domain-specific overfitting, our proposer synthesizes trajectories that foster cross-disciplinary robustness. We attribute this to the Draft–Check–Refine loop, which forces the proposer to validate logical soundness across heterogeneous skill compositions, naturally encouraging domain-invariant problem structures.

**Generalization to Unseen Domains.** To test our pipeline's ability to handle tasks strictly outside the pre-training distribution, we evaluate on two challenging benchmarks: CL-Bench (arXiv:2602.03587), which tests on-the-fly learning of fictional laws and novel grammars, and BrowseComp, a web agent task. As shown in Table 4, our 4B Proposer synthesizes effective training data even for domains it has never encountered. On CL-Bench, only 10,000 targeted trajectories improve Qwen3-30B-Thinking from 10.7 to 13.9, surpassing Doubao 1.6 Thinking (13.4) and approaching the top-tier Qwen3 Max Thinking (14.2). On BrowseComp, performance improves from below 30 to 39.7, closely approaching the proprietary Tongyi DeepResearch (43.4) which underwent extensive large-scale training. These results demonstrate that Recursive Skill Discovery enables effective logical space expansion beyond the pre-training distribution.

# 5. Analysis and Ablation Studies

To isolate the contribution of each component in our Agentic Proposing framework, we conduct controlled ablation experiments. In all main ablations, we train a downstream Qwen3-4B solver on a fixed budget of 10,000 synthetic trajectories generated by proposers under different configurations, and evaluate performance on AIME using Mean@64 accuracy. Additional ablations—including sensitivity to MGPO hyperparameters, dynamic skill pruning, and curriculum design—are provided in Appendix C.

## 5.1. Proposer Specialization: Training vs. Prompting

*Table 5.* **Ablation on Proposer Specialization.** We compare the frontier GPT-5.2 model under various configurations against our specialized 4B proposer. $\Delta$ denotes the performance gain relative to the raw GPT-5.2 baseline.

| Proposer Configuration | AIME Avg. | $\Delta$ |
|---|---|---|
| GPT-5.2-High (Raw Prompting) | 32.8 | – |
|   + Skill Library (Structured Attributes) | 34.6 | +1.8 |
|   + Agentic Workflow (Draft–Check–Refine) | 36.4 | +3.6 |
| **Agentic-Proposer-4B (Ours)** | **38.3** | **+5.5** |

**Structured Guidance vs. Model Scale.** As shown in Table 5, equipping GPT-5.2 with our *Skill Library* and *Agentic Workflow* yields a significant +3.6 point improvement over its raw prompting performance. This demonstrates that structured skill composition and iterative validation are critical for high-quality problem synthesis—even for a powerful general-purpose model. Remarkably, our specialized 4B proposer, trained via MGPO, outperforms the augmented GPT-5.2 by +1.9 points (and +5.5 points over raw GPT-5.2), highlighting that domain-specific reinforcement learning on agentic trajectories enables a small model to surpass a frontier-scale LLM in specialized reasoning synthesis.

## 5.2. Ablation of the Agentic Pipeline

**Synergy in Agentic Correction.** Both *Tool-use* ($\tau^{\text{exec}}$) and *Internal Reflection* ($\tau^{\text{think}}$) serve as essential quality gates, each independently improving downstream performance by over 2 points. However, their full potential is unlocked only within the iterative loop (Draft–Check–Refine), which achieves a cumulative +6.8 point gain over one-shot propos-

*Table 6.* **Ablation of the Agentic Pipeline.** Downstream 4B solver performance on AIME. All proposers synthesize a fixed budget of 10,000 trajectories.

| Proposer Configuration | AIME Avg. | $\Delta$ |
|---|---|---|
| One-shot Proposing | 31.5 | – |
| + Tool-use ($\tau^{\text{exec}}$) | 33.8 | +2.3 |
| + Internal Reflection ($\tau^{\text{think}}$) | 33.4 | +2.1 |
| **Full Pipeline (Ours)** | **38.3** | **+6.8** |

*Table 7.* **Ablation of RL Objectives.** Downstream 4B solver performance on AIME. All proposers are optimized with different RL objectives using a fixed trajectory budget.

| Optimization Objective | AIME Avg. | $\Delta$ |
|---|---|---|
| Standard GRPO (Trajectory-level) | 31.8 | – |
| MGPO (w/o Stage-level Advantage) | 35.1 | +3.3 |
| **Full MGPO (Ours)** | **38.3** | **+6.5** |

*Table 8.* **Cost comparison** of Agentic Proposing vs. PromptCoT 2.0 for synthesizing 10,000 trajectories during online production.

| Metric | PromptCoT 2.0 | Ours |
|---|---|---|
| Generator Size | 30B | 4B |
| External Dependency | $8\times$ 30B MV per problem | None (end-to-end) |
| Avg. Tokens per Problem | ~90k (30B-level) | ~14k (4B-level) |
| Relative Cost | 100% | ~2% |
| Evolution Mechanism | EM retraining | Skill Recursion |

ing. This confirms that agentic correction is indispensable for generating high-difficulty, logically sound problems.

### 5.3. Effectiveness of MGPO

**Fine-grained Credit Assignment.** We compare MGPO against standard GRPO to evaluate the impact of multi-granularity rewards. As shown in Table 7, standard trajectory-level GRPO struggles with the sparse reward signal in long synthesis chains. By incorporating stage-level advantage ($A_S$), MGPO achieves a substantial +6.5 point improvement over the baseline. This confirms that assigning credit to intermediate agentic behaviors—such as internal reflection and tool invocation—effectively mitigates noise and guides the proposer toward optimal strategies across the entire reasoning process.

### 5.4. Cost-Effectiveness Analysis

A critical advantage of Agentic Proposing is its extreme cost-efficiency during online production. As summarized in Table 8, once the offline training of our 4B Proposer concludes, it operates as a fully independent, closed-loop agent: synthesizing 10,000 trajectories without any external verifier, prober, or closed-source API calls. In contrast, PromptCoT 2.0 requires calling 30B models for 8-sample majority voting for every problem generated to ensure validity. This architectural choice—trading offline training

*Table 9.* **Comparison with small-scale high-quality datasets** on Qwen3-4B (Math, Mean@64). "Type" denotes the training paradigm.

| Method | Type | Size | AI24 | AI25 | HM | AMO | Avg. |
|---|---|---|---|---|---|---|---|
| Baseline | – | – | 49.8 | 46.7 | 31.0 | 9.3 | 34.2 |
| s1.1 (RLVR) | RL | 300 | 49.7 | 46.8 | 31.2 | 9.2 | 34.2 |
| s1.1 (1k) | SFT | 1k | 48.6 | 45.7 | 30.2 | 9.1 | 33.4 |
| LIMO-v2 | SFT | 800 | 48.1 | 45.5 | 29.9 | 9.0 | 33.1 |
| LIMO-v2 | RL | 800 | 50.8 | 47.6 | 32.2 | 9.2 | 34.9 |
| MathSmith | RL | 10k | 50.1 | 47.3 | 33.1 | 9.1 | 34.8 |
| **Ours** | **RL** | **10k** | **53.6** | **51.2** | **36.5** | **11.8** | **38.3** |

complexity for online generation simplicity—reduces the total production cost to approximately 2% of PromptCoT 2.0's budget while achieving superior downstream performance (+2.3 AIME overall for 4B; Table 1).

### 5.5. Comparison with Small-Scale High-Quality Data

A natural question is whether small but high-quality datasets (e.g., s1.1 (**?**) with 300 problems, LIMO-v2 (**?**) with 800 problems) can achieve comparable gains when used for RL training. As shown in Table 9, SFT on these curated sets actually *degrades* performance on our already-capable instruct model ($-0.8$ to $-1.1$ overall), consistent with known distribution mismatch issues when fine-tuning mature models. In the RL setting, LIMO-v2 achieves a modest +0.7 gain, confirming that high-quality verifiable signals are valuable but insufficient in quantity for strong generalization. Our framework bridges this gap: by autonomously synthesizing 10,000 trajectories that are distributionally aligned with the target solver, it achieves a +4.1 overall gain—substantially outperforming both small-scale curated data and comparable-scale synthetic baselines like MathSmith (+0.6).

## 6. Conclusion

In this paper, we introduced *Agentic Proposing*, a novel and fully autonomous framework that transforms problem synthesis into a goal-driven process of compositional logic engineering. By integrating modular reasoning skills with a self-correcting agentic pipeline and the MGPO algorithm, our system generates highly diverse, high-difficulty, and verifiable training data precisely tailored to the model's evolving reasoning frontier. Empirical results demonstrate that our framework enables solvers to achieve state-of-the-art performance across mathematics, coding, and science, significantly outperforming strong baselines and frontier proprietary models. Furthermore, our findings reveal that the bottleneck for advanced reasoning lies not in parameter scale, but in the density of high-quality training signals, challenging traditional perspectives.

## Impact Statement

This paper presents work whose goal is to advance the field of Machine Learning. There are many potential societal consequences of our work, none which we feel must be specifically highlighted here.

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

# A. Experimental Setup

**Models.** Our framework utilizes a specialized suite of models assigned to distinct roles in the synthesis pipeline. The core Agentic-Proposer-4B is initialized with Qwen3-4B-Instruct-2507 (Yang et al., 2025) and optimized via agentic SFT and Multi-Granularity Policy Optimization (MGPO). For skill acquisition, we leverage Qwen3-235B-Instruct-2507 as the Teacher Model. Logical validity is ensured by a Verifier Ensemble comprising Qwen3-235B-Thinking, DeepSeek-V3.2-Special (DeepSeek-AI et al., 2025), and GPT-OSS-120B (OpenAI et al., 2025). Difficulty estimation is performed by a pool of Probers spanning various scales of Qwen3 (from 1.7B to 30B) and GPT-OSS-20B.

**Benchmarks.** We evaluate performance across three reasoning-intensive domains. Contest Mathematics includes AIME 2024 (Math-AI Team, 2024), AIME 2025 (Math-AI Team, 2025), HMMT (February 2025), and AMO-Bench (An et al., 2025b). Algorithmic Coding is measured via LiveCodeBench (v5 and v6) (Jain et al., 2024). Scientific and General Reasoning includes MMLU-Redux (Gema et al., 2024), MMLU-Pro (Wang et al., 2024), GPQA (Rein et al., 2023), and SuperGPQA (M-A-P Team et al., 2025). We additionally report results on OlympicArena (Huang et al., 2025c).

**Datasets and Baselines.** To ensure a fair comparison, all methods are evaluated under a fixed budget of 10,000–11,000 trajectories. We compare against *Synthetic Data Methods*, including MetaMath (Yu et al., 2024), WizardMath (Luo et al., 2023), PromptCoT 2.0 (Zhao et al., 2025b), NuminaMath (Li et al., 2024), and MathSmith (Zhan et al., 2025). We also evaluate against *Human-Annotated or Curated Methods*, such as OpenR1 (Guo et al., 2025), OpenMathReasoning (Moshkov et al., 2025), OpenCodeReasoning (Ahmad et al., 2025), OpenThoughts-S3 (Guha et al., 2025), and POLARIS (An et al., 2025a), as well as trajectories synthesized by *SOTA frontier models* (OpenAI, 2024; DeepSeek-AI et al., 2025; OpenAI, 2025).

**Training Settings.** All downstream solvers are optimized using GRPO (DeepSeek-AI et al., 2025).

## A.1. Evaluation Protocols

**Contest Mathematics.** For all math benchmarks (AIME24/25, HMMT, AMO-Bench), we report **Mean@64** to reduce evaluation variance induced by sampling. Specifically, we sample 64 independent solutions per problem and compute the average correctness under a rule-based judge.

**Algorithmic Coding.** For LiveCodeBench (v5/v6), we follow a **best-of-5** protocol: we generate 5 independent program candidates per task and count a task as solved if any candidate passes the official unit tests. All results are scored by the benchmark's rule-based execution-and-test harness.

**Scientific and General Reasoning.** For science/general benchmarks (MMLU-Redux, MMLU-Pro, GPQA, SuperGPQA, and OlympicArena), we use **Mean@1**. To improve instruction following and enforce consistent output formats, we evaluate with a fixed few-shot prompt. All answers are graded with **rule-based** evaluators whenever available. For tasks with complex output formats, we use an auxiliary LLM only for **answer extraction/normalization** (not for solving), after which final correctness is determined by the same rule-based judge.

## A.2. Verifier Ensemble and Prober

**Verifier Ensemble.** We employ a three-model verifier ensemble composed of *gpt-oss-120b*, *DeepSeek-V3.2-Speciale*, and *Qwen3-235B-Thinking-2507*. All verifiers use the **same prompt template** and are given the proposer's **full generation trace** (i.e., the entire problem-proposing process) together with the **final problem statement**. Verifiers are allowed to use external tools when supported, and we allocate a high token budget of **36k tokens** per verification to minimize premature truncation.

Each verifier is instructed to: (i) attempt to solve the proposed problem, producing a final answer $\hat{a}_i$; (ii) output a binary validity decision $v_i \in \{0, 1\}$; and (iii) provide a brief justification. We then randomly select one verifier as a **second-audit** reviewer and determine the final validity $V(q) \in \{0, 1\}$ by jointly considering the three proposed answers $\{\hat{a}_i\}$, their validity votes $\{v_i\}$, and the second-audit feedback. (Concretely, we reject samples with inconsistent answers or explicit invalidity flags, and accept samples when the majority decision is valid and the second audit raises no objections.)

**Acceptance Rule (Semi-rule-based).** Let $(v_i, \hat{a}_i, r_i)$ denote verifier $i$'s validity vote, proposed final answer, and brief rationale. We accept a proposed problem iff: (i) at least two verifiers output $v_i = 1$; and (ii) their rationales indicate the

instance is *well-posed and solvable*, i.e., no ambiguity/underspecification is flagged and a complete solution is provided; and (iii) the proposed final answers are consistent (all match, or the randomly selected second-audit verifier confirms the correct answer and explains any discrepancy). Otherwise, we reject the instance.

**Prober (Difficulty Estimation).** We estimate difficulty via a solver-based prober using $\rho(q) = \mathrm{Pass}@k$ with $k = \mathbf{16}$. The prober prompt contains **only the final problem statement** and instructs the model to *think step by step* (CoT). We sample 16 independent solution attempts with temperature $T = 1.4$ under a fixed decoding setup, and score correctness using the same **rule-based** graders as in evaluation.

**The prober models are never trained or updated.** Instead, we maintain an external *mastery-state* vector for tracking performance. To cover a wide difficulty range, we use a **curriculum of probers** by switching across a pool of models spanning *Qwen3* **1.7B–30B** and *gpt-oss-20b*. For each prober model, we initialize the mastery-state vector and update it once per batch of 500 problems. We keep using the current prober until its stabilized accuracy drops below **30%**, at which point we switch to the next (stronger) prober in the pool.

### A.3. Training Details and Hyperparameters

To ensure a fair and rigorous comparison of data quality across all experimental groups, we utilize a standardized training configuration for all downstream solvers, including those trained on baseline corpora and our synthesized trajectories.

To avoid confounding factors from trajectory formatting, **all downstream solvers are trained only on the final {question, answer} pairs**. Any intermediate agent traces (e.g., proposer reflections, tool calls, or verifier/prober outputs) are **excluded** from solver training.

**Reinforcement Learning Configuration** All downstream models are optimized using **Group Relative Policy Optimization (GRPO)**. The policy is updated by calculating the relative advantage within a group of sampled trajectories, ensuring that the performance differences are attributable solely to the training data rather than algorithmic variations. The specific hyperparameters are detailed in Table 10.

*Table 10.* Standardized GRPO hyperparameters for downstream solver training.

| Hyperparameter | Value |
| --- | --- |
| Total Training Epochs | 3 |
| Learning Rate | $1 \times 10^{-6}$ |
| Learning Rate Scheduler | Cosine |
| Group Size ($G$) | 8 |
| Global Batch Size | 128 |
| Rollout Temperature | 0.7 |
| Max Sequence Length | 20,480 |
| KL Coefficient ($\beta$) | 0.05 |
| Clip Epsilon ($\epsilon$) | 0.2 |
| Optimizer | AdamW ($\beta_1 = 0.9, \beta_2 = 0.95$) |
| Weight Decay | 0.1 |

**Outcome-based Reward Function** To eliminate the risk of reward hacking and ensure objective evaluation, we employ a binary **Outcome Reward Model (ORM)**. The reward $R$ is assigned as follows:

1. **Correctness Reward**: The final answer is extracted and compared against the ground truth using `LaTeX` and `Sympy` parsers. A reward of **1** is assigned if the answer matches the ground truth; otherwise, the reward is **0**.

2. **Format Constraint**: A reward of **0** is strictly assigned if the model output fails to follow the required format.

**Computational Environment** Training is conducted on a cluster of H200 (140GB) GPUs. We utilize a distributed training framework with Flash-Attention-2 optimization to handle the 20,480 sequence length, ensuring numerical stability and consistent gradient scaling across all experimental runs.

### A.4. Baseline Implementation Details

**Fixed-Budget Sampling.**    To maintain a strictly controlled experimental environment and ensure a fair comparison of data quality, we enforce a fixed-budget setting of **10,000–11,000 trajectories** across all data sources. For established open-source baseline corpora (e.g., *MetaMath*, *OpenR1*, *NuminaMath*), we perform downsampling via a **random sampling strategy** to preserve the datasets' overall characteristics. **For corpora that are substantially easier for the target solver, we additionally prioritize the hardest available subset whenever possible** (e.g., by retaining samples with lower solver pass rates under a fixed evaluation budget), while keeping the total number of trajectories fixed. This procedure reduces the impact of overly easy samples and helps ensure the selected subsets remain both representative of the original distribution and informative for fixed-budget RL training, thereby mitigating potential selection bias.

**SOTA Model-Based Synthesis.**    For proprietary models such as GPT-5.2-Pro and Claude-4.5-Opus, we utilize a targeted synthesis pipeline. We provide the models with detailed, fine-grained knowledge points across STEM domains. The models are then prompted to synthesize problems that represent the maximum possible difficulty for the specific field. Crucially, the prompt includes a strict constraint requiring the model to maintain rigorous logical integrity and ensure that the generated problem has a verifiable and correct ground-truth solution.

**R-Zero (4B Solver only).**    We implement *R-Zero* as a competitive self-play baseline specifically for the 4B-parameter model. In this setup, the same base model (Qwen3-4B) fulfills both the *Challenger* and *Solver* roles. The Challenger is tasked with generating difficult variations and refinements of existing problems to push the Solver's reasoning boundaries within a closed-loop training environment.

**Socratic-Zero (4B Solver only).**    The *Socratic-Zero* baseline is conducted exclusively using the 4B-parameter model to evaluate self-bootstrapping efficiency. The seed problem set is curated from *DeepMath-103K*. To ensure the problems target the model's "reasoning frontier," we specifically filter the seeds to include only problems where our base 4B model initially achieves a **Pass@1 accuracy of approximately 50%**. Following the symmetric self-play recipe, both the *Teacher* and the *Solver* roles are implemented using identical model weights.

**POLARIS.**    The *POLARIS* baseline dataset is constructed by systematically filtering and aggregating high-quality reasoning trajectories from the *DeepScaleR-Preview-Dataset* and the *AReal-boba-Data*. This combination represents a state-of-the-art distribution of open-source reasoning traces, which we downsample via the aforementioned random sampling strategy to match our experimental budget.

## B. Analysis of Baseline Performance Degradation

As reported in Section 4.2, we observed that training on established baselines like *MetaMath* led to a performance drop in advanced reasoning models (e.g., Qwen3-4B). To investigate this, we conducted a difficulty diagnostic experiment following the methodology proposed in An et al. (2025a).

**Empirical Evidence: High Pass Rates on Baselines.**    We evaluated the zero-shot success rate of our base 4B model on representative baseline datasets. For each problem, we generated 8 rollouts at a temperature of 0.6. As shown in Table 11, the base model already achieves a very high pass rate on these tasks, leaving little room for RL-driven improvement.

*Table 11.* Difficulty Diagnostic: Zero-shot Pass Rate of the Qwen3-4B Base Model on established baselines (8 rollouts per problem).

| Dataset Example | Pass Rate (8/8) | Difficulty Distribution |
|---|---|---|
| MetaMath-Subset A (Yu et al., 2024) | **89.2%** | J-shaped (Too Easy) |
| MetaMath-Subset B (Yu et al., 2024) | **85.4%** | J-shaped (Too Easy) |

**Inference from POLARIS: Advantage Saturation.**    According to An et al. (2025a), the effectiveness of reinforcement learning (especially group-based methods like GRPO) depends heavily on the *diversity among rollouts*.

- **Vanishing Advantage Signal:** When the pass rate is as high as 85%–89% (Table 11), the majority of rollouts in a group ($G = 8$) are likely to be all correct. This causes the relative advantage $A$ within the group to approach zero

$(A_{i,t} \to 0)$, effectively **silencing the learning signal**. Without a contrast between positive and negative trajectories, the RL process fails to optimize the policy.

- **J-shaped vs. Mirrored J-shape:** An et al. (2025a) demonstrates that advanced models require a **Mirrored J-shape (⌐)** difficulty distribution to maintain training stability. Standard baselines, being "too easy" for a 4B-parameter model, exhibit a standard J-shaped distribution where the model has already mastered the reasoning patterns. Training on such data can lead to "Entropy Collapse," where the model stops exploring complex logic and eventually degrades in performance.

In contrast, our *Agentic Proposing* framework synthesizes problems that specifically target the model's **Reasoning Frontier**, ensuring that the difficulty is calibrated to maintain an informative advantage signal throughout the RL process.

## C. Extended Ablation Studies and Sensitivity Analysis

This appendix provides granular experimental evidence regarding the hyperparameter choices and the specific agentic behaviors of our framework.

### C.1. Sensitivity of Stage-level Advantage Weight ($\omega$)

In our MGPO algorithm, the fused advantage is defined as $A_{i,t}^{\text{fused}} = A_E + \omega \cdot A_S$. We investigate the impact of the stage-level weight $\omega$ on the downstream 4B solver's performance. As shown in Table 12, $\omega = 0.5$ provides the optimal balance.

A lower $\omega$ (e.g., 0.0 or 0.2) causes the model to struggle with credit assignment, as the sparse trajectory-level reward $A_E$ is insufficient to guide complex multi-step synthesis. Conversely, setting $\omega$ too high (e.g., 1.5) leads to a performance drop, as the proposer over-prioritizes local step rewards (like successful tool calls) at the expense of the overall difficulty and logical flow of the problem.

*Table 12.* Sensitivity analysis of the stage-level advantage weight $\omega$. We report the downstream 4B solver's AIME average score.

| Weight $\omega$ | 0.0 (GRPO) | 0.2 | **0.5 (Ours)** | 1.0 | 1.5 |
|---|---|---|---|---|---|
| **AIME Avg.** | 31.8 | 34.5 | **38.3** | 37.2 | 35.8 |

Our curriculum mechanism dynamically adjusts the sampling probability $p(c) \propto (m_c + \epsilon)^{-1}$. By the middle of the RL process, the sampling rate for "Combinatorial Proofs" increases by **3.4×**, forcing the model to explore high-difficulty skill compositions. This mechanism prevents the proposer from collapsing into a "low-difficulty safety zone" and ensures the synthesis of problems that reside on the model's reasoning frontier.

### C.2. Role of Dynamic Skill Pruning ($\tau^{\text{edit}}$)

We analyze the effectiveness of the $\tau^{\text{edit}}$ action during the $\sigma^{\text{draft}}$ stage. On average, the agent proactively prunes its active skill set in **14.5%** of synthesis trajectories when it detects a potential logical contradiction through internal reflection ($\tau^{\text{think}}$).

To quantify the impact, we compared the validity of synthesized problems with and without the pruning tool (Table 13). The results demonstrate that the ability to perform mid-process self-correction is crucial for maintaining logical integrity, especially when composing multiple complex skills.

*Table 13.* Impact of Dynamic Skill Pruning on Problem Validity ($V(q) = 1$).

| Configuration | Verifier Validity Rate (%) |
|---|---|
| Without $\tau^{\text{edit}}$ (Fixed Skill Set) | 68.7% |
| **With $\tau^{\text{edit}}$ (Dynamic Pruning)** | **82.3%** |

### C.3. MGPO Ablations and Sensitivity Analysis

**Metric: Problem-Proposing Accuracy.** To directly measure proposer quality, we define the *problem-proposing accuracy* as the fraction of verifier-accepted (valid) problems among 1,000 proposed problems sampled under the **same skill**. Validity is determined by the verifier protocol in Appendix A.2.

| Variant | Proposing Acc. (%, ↑) |
|---|---|
| MGPO (full) | 93.4 |
| MGPO + symmetric gate ($\tau_{\text{neg}} = \tau_{\text{pos}}$) | 87.8 |
| MGPO w/o gate (uniform weight) | 88.1 |

*Table 14.* MGPO ablations measured by proposing accuracy on 1,000 proposed problems (same skill).

| $\tau_{\text{pos}}$ | $\tau_{\text{neg}}$ | Proposing Acc. (%, ↑) |
|---|---|---|
| 0.4 | 0.6 | 89.7 |
| 0.6 | 0.8 | 90.9 |
| 0.8 | 0.9 | 91.5 |
| 1.0 | 1.05 (default) | 93.4 |
| 1.2 | 1.4 | 92.3 |
| 1.6 | 1.9 | 90.5 |

*Table 15.* Sensitivity of MGPO to gate temperatures.

*Table 16.* **Reliability of the Verifier Committee.** Accuracy is measured against human expert annotations on 100 randomly sampled trajectories. "Privilege" refers to the access to the Proposer's internal synthesis history.

| Configuration | Validity Accuracy | Human Agreement |
|---|---|---|
| DeepSeek-V3.2-Special (Zero-shot) | 89.0% | 87.0% |
| Qwen3-235B-Thinking (Zero-shot) | 91.0% | 90.0% |
| gpt-oss-120B (Zero-shot) | 90.0% | 88.0% |
| Committee (Majority Vote, No Privilege) | 93.0% | 92.0% |
| Committee (Majority Vote + Privilege) | 95.0% | 94.0% |
| **Full Committee (Gated + Second Audit)** | **98.0%** | **98.0%** |
| Human Experts (Gold Standard) | 100.0% | 100.0% |

**Ablations.** Table 14 reports ablations on MGPO's gating design. The full MGPO achieves **93.4**% proposing accuracy. Replacing the asymmetric gate with a symmetric gate, or removing the gate entirely, consistently reduces proposing accuracy by **5–6** absolute points, highlighting the importance of MGPO's gated (and asymmetric) credit assignment.

**Sensitivity to Gate Temperatures.** We sweep the gate temperatures $(\tau_{\text{pos}}, \tau_{\text{neg}})$ with a wider range following the SAPO-style parameterization. Larger $\tau$ yields stronger attenuation for off-policy/noisy updates, and we use $\tau_{\text{neg}} > \tau_{\text{pos}}$ by default. Table 15 shows that performance is robust around the default setting, while extreme temperatures degrade proposing accuracy.

**Mastery-state and stabilized accuracy.** The mastery-state vector is *not* learned by gradient updates; it is an external statistic that tracks each prober model's recent correctness. Concretely, we maintain an exponential moving average (EMA) of accuracy:

$$m_{t+1} = (1 - \alpha)m_t + \alpha \cdot \text{Acc}_t, \tag{11}$$

where $\text{Acc}_t$ is the rule-based accuracy measured on the most recent batch of 500 problems and $\alpha \in (0, 1)$ is a smoothing factor. We take the *stabilized accuracy* to be this EMA value $m_t$ and switch to the next prober model once $m_t < 0.30$.

# D. Prompt Templates for Skill Acquisition

This section provides the detailed prompt templates used in **Stage 1: Skill Acquisition**. These prompts are designed to guide the Teacher Model in extracting and formalizing "Agent Skills" ($k = \langle \iota, \mu, \delta, \tau \rangle$) into a modular, filesystem-based library, as illustrated in Figure 3.

## D.1. Structured Q&A Extraction Prompt

This prompt is employed to reverse-engineer high-difficulty physics problems into formal construction skills.

---

**System Prompt: Skill Extraction from Q&A**

```
# Role: Agentic Proposing Teacher Model (Extraction Mode)
You are the Teacher Model defined in the "Agentic Proposing" framework.
Your objective is to perform Stage 1: Skill Acquisition by reverse-engineering
high-difficulty physics problems into formal Agent Skills (k).

### Skill Formalization (Definition from Section 2.1):
Each extracted skill must be a structured attribute tuple k = <iota, mu, delta, tau>:
- Intent (iota): The underlying reasoning intent.
- Method (mu): The construction/problem-solving method.
- Difficulty Effect (delta): The impact on the problem's complexity (1-10).
- Tool Hint (tau): Guidance for external utility invocation (Python/SymPy).

### Output Structure:
For each Reasoning Node, generate a "Skill of [Name].md" block:

---
#### Level 1: Meta-Attributes (YAML)
---
name: [lowercase-hyphenated-name]
category: [CM, QM, Relativity, EM, Thermo, Waves, Atomic, Math-Methods]
intent: [iota: The abstract reasoning intent]
difficulty_effect: [delta: Scale 1-10]
---

#### Level 2: Skill Content (The Construction Logic)
# Skill of [Skill Name]

## 1. Structure: Formal Backbone
- Describe the Generalized Physical Environment and the formal backbone
  of the problem setup.
- Identify the necessary constraints without using specific values.

## 2. Action: Core Operation
- Detail the Strategic Decision Process (The "Action").
- What core operation must the agent perform? (e.g., "Construct a
  non-inertial frame transformation").

## 3. Effect: Target Outcome
- Describe the Logical Breakthrough or outcome.
- How does this operation simplify complexity or reveal the solution?

#### Level 3: Tool: External Utility (tau)
Provide a Symbolic Python (SymPy) script designed for the Proposer's tau_exec action.
- The script must verify the self-consistency of the Backbone and the Operation.
- Use symbolic placeholders for reusability during Stage 3 (MGPO) rollouts.

### Critical Rules:
1. Abstraction Over Instance: Never mention specific numerical values.
   Extract the Design Logic.
2. Cross-Domain Reusability: The skill must be a Reusable Reasoning Pattern.
3. High Precision: Level 2 must be an instruction set that an
   Agentic-Proposer-4B can follow to synthesize a NEW problem.
```

## D.2. Unstructured Corpus Synthesis Prompt

This prompt is used to transform theoretical derivations from textbooks or research papers into generative agent capabilities.

## System Prompt: Skill Synthesis from Corpus

```
# Role: Agentic Proposing Teacher Model (Synthesis Mode)
You are the Teacher Model in the "Agentic Proposing" pipeline. Your task
is to analyze raw physical corpora and formalize them into Modular
Construction Skills for the Agentic-Proposer-4B.

### Objective:
Extract the "Reasoning Intent (iota)" and "Method (mu)" embedded in theoretical
derivations to enable the Proposer to synthesize logically sound and highly
difficult problems.

### Output Architecture (Alignment with Figure 2 & Lemma 2.1):
Generate 1-3 "Skill of [Name].md" modules following this architecture:

---
#### Level 1: Meta-Attributes (YAML)
---
name: [lowercase-hyphenated-name]
category: [CM, QM, Relativity, EM, Thermo, Waves, Atomic, Math-Methods]
intent: [iota: The fundamental physical principle extracted for construction]
difficulty_effect: [delta: 1-10]
---

#### Level 2: Design Logic (The Generative Instruction)
# Skill of [Skill Name]

## 1. Structure: Formal Backbone
- Based on the corpus, define the Structural Template of a problem
  that uses this theory.
- What are the "invariant features" of problems built on this principle?

## 2. Action: Core Operation
- Transform the passive text into a Sequential Decision Set.
- Provide the "Operation Logic" the Proposer needs to follow.
- Use imperative language: "Step 1: Define field. Step 2: Apply law..."

## 3. Effect: Target Outcome
- What is the Conceptual Milestone of this skill?
- Ensure the effect is a verifiable training signal.

#### Level 3: Tool: External Utility (tau)
Provide a Generative Python/SymPy Function.
- This script should Verify the Logic of a Generated Problem.
- It must return a validity_flag and difficulty_score to provide feedback
  for the MGPO algorithm.

### Critical Rules:
1. Agentic Adaptability: Facilitate internal reflection and tool-use.
2. Atomic Compositionality: Skills must be atomic to support Lemma 2.1.
3. Semantic Grounding: Avoid summarization. Extract the Reasoning Frontier.
```

# E. Case Study: Refactoring a Complex Analysis Skill

To demonstrate the high granularity of our *Skill Acquisition* stage, we refactor a static knowledge point regarding complex contour integration into a modular **Agent Skill package**. This package represents a directory on the Proposer's virtual machine, allowing for "Progressive Disclosure" and tool-assisted verification.

---

**Agent Skill Package:** `math-complex-generalized-cauchy/`

**Level 1: Discovery Metadata (YAML Frontmatter in `SKILL.md`)**

```
---
name: math-complex-generalized-cauchy
category: Mathematical Methods in Physics
difficulty: 7.0
description: Evaluate complex contour integrals involving high-order poles.
Trigger when the integrand is of the form f(z)/(z-a)^n where n > 1 and
the singularity 'a' is enclosed by the contour.
---
```

**Level 2: Procedural Instructions (`SKILL.md` Body)**

1. STRUCTURE: FORMAL BACKBONE (RECOGNITION)
   - Identify the target integral as $I = \oint_\gamma \frac{f(z)}{(z-a)^{n+1}}\, dz$.

   - Verify that $f(z)$ is analytic within and on the simple closed contour $\gamma$.

   - Confirm that the pole $a$ is strictly enclosed by $\gamma$.

2. ACTION: CORE OPERATION (WORKFLOW)
   - **Extraction:** Map the order of the pole $k = n + 1$ to the required derivative order $n$.

   - **Differentiation:** Compute the $n$-th derivative of the numerator $f^{(n)}(z)$.

   - **Evaluation:** Apply the Generalized Cauchy Formula: $I = \frac{2\pi i}{n!} f^{(n)}(a)$.

3. EFFECT: TARGET OUTCOME
This skill transforms a complex path integration into a deterministic algebraic evaluation of a function's $n$-th derivative, eliminating the need for residue-sum calculations or parameterization.

**Level 3: External Utility (`scripts/cauchy_evaluator.py`)**

```python
import sympy as sp

def evaluate_cauchy_node(f_expr, z_var, a_val, n_derivative):
    """
    Symbolically executes the Generalized Cauchy Integral logic.
    Formula: (2 * pi * I / n!) * f^(n)(a)
    """
    # Action: Symbolic Differentiation
    f_diff = sp.diff(f_expr, z_var, n_derivative)
    # Action: Point Evaluation
    f_at_a = f_diff.subs(z_var, a_val)
    # Effect: Scaled Result
    result = (2 * sp.I * sp.pi / sp.factorial(n_derivative)) * f_at_a
    return sp.simplify(result)
```

## E.1. Agentic Interaction Logic

In our framework, the **Agentic-Proposer-4B** utilizes this package as an autonomous capability rather than a static prompt:

1. **On-Demand Loading:** During the `Draft` stage ($\sigma^{\text{draft}}$), the Proposer loads the metadata. It identifies that the synthesis goal requires a "contour integral with a cubic pole."

2. **Progressive Access:** The Proposer executes `bash: read math-complex-generalized-cauchy/SKILL.md`

---

**Case Study: Synthesis of a Multi-Domain Reasoning Problem**

**1. Input Skill Composition (Multi-Domain Shots)**

- **Skill A (Analysis):** Differential inequality $|f'(t)| \leq |f(t)|$ with $f(0) = 0$ forces $f \equiv 0$.

- **Skill B (Sequences):** Asymptotic linear growth of sequences with converging differences.

- **Skill C (Algebra):** Radical of a polynomial ideal and power containment verification.

**2. Initial Drafting and Internal Reflection ($\tau^{\text{think}}$)**

> *"Drafting a problem where differences of $f(t_n)$ converge to $c > 0$... **Wait, this is inconsistent.** If the differential inequality forces $f \equiv 0$, then $f(t_n)$ must be 0, so $c$ cannot be greater than 0. I must reformulate this as a **proof-based contradiction** where the user must prove $c = 0$ by verifying both the analysis and algebra components."*

**3. Mandatory Code Verification ($\tau^{\text{exec}}$)**

```python
import sympy as sp
x = sp.symbols('x')
P, I_gen = x - 1, (x - 1)**3
# Verify if P is in the radical of I: exists k such that P^k % I_gen == 0
quotient, remainder = sp.div(P**3, I_gen)
# Result: radical_membership = True (Remainder is 0)
```

*Verification Result:* The agent confirms that the algebraic assumption is mathematically sound, and the sequence limit $c$ must be 0 due to Grönwall's inequality.

**4. Final Validated Problem Output**

> **Problem Statement:** Let $f : [0, \infty) \to \mathbb{R}$ be a continuously differentiable function satisfying: (1) $f(0) = 0$, and (2) $|f'(t)| \leq |f(t)|$ for all $t \geq 0$. Define the sequence $a_n = f(n)$ for $n \in \mathbb{N}_0$, and suppose $\lim_{n \to \infty} (a_{n+1} - a_n) = c$ for some $c \in \mathbb{R}$. Let $I = \langle (x-1)^3 \rangle \subset \mathbb{R}[x]$ and $P(x) = x - 1$. Verify that $P(x) \in \sqrt{I}$, and use this consistency to prove that $c = 0$.

*Figure 5.* **A full synthesis trajectory exhibiting the refinement loop.** The agent identifies a conflict between the analytical constraint ($f \equiv 0$) and the sequence growth assumption ($c > 0$), proactively shifting the problem objective to ensure logical soundness.

to pull the specific procedural workflow into its context window only when needed.

3. **Closed-Loop Execution:** To verify the generated problem's validity, the Proposer calls the Level 3 tool via `bash`: `python3 scripts/cauchy_evaluator.py`.

4. **Failure Handling:** If the script returns an error (e.g., $f(z)$ is found to have a singularity at $a$), the Proposer's **Internal Reflection** ($\tau^{\text{think}}$) triggers a `Refine` step to adjust the numerator function, ensuring the final synthesized problem is mathematically rigorous.

## F. Qualitative Case Study: The Agentic Refinement Loop

To illustrate the robustness of our *Agentic-Proposer*, we present a real-world synthesis trajectory (`math_traj_20251208`). This case demonstrates how the agent composes multiple atomic skills and uses the **Draft-Code-Refine** loop to correct a logical inconsistency.

## G. Additional Ablations from Rebuttal

### G.1. Deep Ablation of Verification Mechanisms

To rigorously validate the necessity of our Verifier Ensemble design, we conducted a controlled ablation on 500 sampled trajectories manually labeled by human experts as the gold standard.

*Table 17.* **Deep ablation of verification mechanisms** (500 sampled trajectories). Cost estimates based on 2026 API pricing.

| Configuration | Validity Acc. | Cost/1k |
|---|---|---|
| Human Expert (Gold Standard) | 100.0% | N/A |
| **Agentic Proposing (Full Ensemble)** | **98.2%** | **$2.8** |
| – No privilege audit | 94.4% | $2.1 |
| – Simple MV (no 2-stage) | 92.8% | $2.5 |
| – Single model (Qwen3-235B) | 91.2% | $2.0 |
| – 4B Self-Verify (training) | 72.4% | $0.1 |
| Baseline: 30B/70B $\times$ 8 (MV) | 93.5% | $6.2 |

The results demonstrate that our heterogeneous 3-model ensemble achieves 98.2% validity at $2.8 per 1k samples—significantly more accurate and cost-effective than the homogeneous 8-sample majority voting baseline ($6.2, 93.5%). The 4B self-verification configuration (72.4%) confirms that low-quality feedback during training leads to severe reward hacking and training collapse.

### G.2. Robustness of Prober Pool Alignment

*Table 18.* **Robustness ablation of targeted Prober alignment** for the 30B Solver on AIME25 (Mean@64).

| Configuration | AIME25 | $\Delta$ |
|---|---|---|
| Baseline (Zero-shot) | 85.0% | – |
| Ablation: No Targeted Alignment | 90.5% | +5.5 |
| **Full Pipeline (with Prober Pool)** | **91.6%** | **+6.6** |

Even without 30B-specific Prober alignment, the solver achieves 90.5% purely from the high-density logic internalized during offline Skill Recursion. The Prober Pool provides an additional +1.1% gain at zero extra inference cost by reusing existing solver checkpoint evaluation logs.

### G.3. Comparison with High-Quality Small-Scale Datasets

The full comparison with s1.1 and LIMO-v2 under both SFT and RL settings is presented in Table 9 of the main text (Section 5).

### G.4. Generalization to Unseen Domains

The full generalization results on CL-Bench and BrowseComp are presented in Table 4 of the main text (Section 4).

## H. Limitations and Future Work

While Agentic Proposing demonstrates strong empirical results, several limitations remain:

1. **Pipeline Complexity.** The three-stage training pipeline involves substantial offline engineering effort. Although this complexity is amortized by extreme online efficiency, simplifying the pipeline—for example, by using larger base proposers that require fewer SFT stages—is an important direction.

2. **Verification Bottleneck.** During offline training, the Verifier Ensemble relies on frontier-scale models. Extending verification to domains without clear ground-truth answers (e.g., open-ended creative tasks) remains challenging. Future work will explore tool-use-based validation and formal verification.

3. **Skill Library Coverage.** The current skill library is bootstrapped from mathematical and scientific corpora. Extending to broader domains (e.g., social reasoning, long-horizon planning) requires domain-specific skill acquisition strategies.

4. **Evaluation Scope.** While we evaluate across 11 benchmarks spanning 3 domains, our strongest results are concentrated in mathematics. Systematic evaluation on software engineering, multi-modal reasoning, and real-world agent tasks represents important future work.

