# OpenReview forum: "Agentic Proposing: Enhancing Large language Model Reasoning via Compositional Skill Synthesis"
_ICML.cc/2026/Conference — ICML 2026 regular_

### Official Review · Reviewer_dybg · 2026-02-25

**Soundness:** 3
**Presentation:** 3
**Significance:** 2
**Originality:** 3
**Overall Recommendation:** 4
**Confidence:** 4

**Summary:**

The focus of this paper is on generating data for reasoning. The main contribution is framing dataset generation as a sequential decision making problem, using a skill-based approach. The method follows three stages
1. first seed problems are used to obtain a set of skills
2. then data is obtained for each skill from a teacher model, and a proposer policy is trained via SFT on the teacher data
3. then RL is applied based on skill proficiency

They generate training data across math, coding, science, and show gains on AIME, HMMT, and AMO-Bench for math, LiveCodeBench for code, and MMLU/GPQA/SuperGPQA/OlympicArena for reasoning, where they show improvements.

**Compliance With Llm Reviewing Policy:**

Affirmed.

**Final Justification:**

The rebuttal has addressed many of my concerns about clarity and I have increased my scored from 3 -> 4

**Key Questions For Authors:**

## Larger Questions
- What is HMMT, where is this dataset from?

## Typos/smaller questions
- L117 missing space
- Figure 1 caption: what do you mean by well-calibrated?
- L089 we introduce
- L in language in title should be capitalized
- small quibble: in Figure 4 why are the bars color-graded?

**Limitations:**

Authors should describe limitations of their work/room for future work, even if it's in the appendix.

**Strengths And Weaknesses:**

# Strengths


- the proposer training pipeline is valuable. MGPO and the reward shaping to encourage difficult but valid problems is a good contribution to the data generation domain
- Their proposer consistently improves over baseline methods, including well-known synthetic datasets like MetaMath, NuminaMath, as well as distilling from problems generated by a single model.
- The method generalizes to code, math, and other reasoning domains like OlympicArena, MMLU, SuperGPQA.



# Weaknesses
- The writing is confusing in places. It's not clear what the input and output of the agent are, even after reading 3.1. The state is "a latent state representing the underlying integrity and difficulty of a problem". What does that mean in practice? How is that latent state obtained? And in the action space, does $\tau^{submit}$ also produce an answer to the problem? Where are the the solutions to generated problems coming from?
- Some other points of confusion for the writing
	- it's not clear to me where the solver enters the loop in Stage 3. Ostensibly, you want the proposer to propose problems that are at the boundary of what the solver can solve. But it's not clear to me where in the training that frontier is measured. Is this via the curriculum in 3.5.1?
	- some of the method seems overengineered, making the description of the method hard to follow
	- The description of MGPO is pretty hard to follow. For example the authors say "addresses the KL-constrained reward maximization problem through a variational reformulation" but they haven't described what that problem actually is, or what how the reformulation is variational.
	- The experiments section doesn't actually say which 4B and 30B models they train as solvers.
	- it's not clear why Lemma 3.1 is needed or why this needs to be a lemma, it feels like a pretty intuitive idea that doesn't need a math wrapper. It would be more accessible if just stated in language.

- The writing anthropomorphizes more than I would like. There's a lot of dicussion of "cognitive" capabilities/"cognitive" actions which I don't think is necessary or warranted.

- Missing related work: treating data generation as a sequential decision making problem was already proposed https://arxiv.org/abs/2410.06215 (ICLR 2025), reducing the novelty of this work.

- I'm not completely convinced about the source of the gains. One difference between the proposed method and the baselines seems to be (though it's hard to tell from the method description) that the generated data is produced by multiple models (in this case, a pipeline of proposer/verifier/prober, where there are ensembles of models for verifiers/probers). While the single-model baselines show decreases in performance in Table 1, I do wonder whether that could be due to a lack of diversity in single-model data. To justify the complexity of the pipeline here, it would be worth examining whether mixed data improves performance (without any agentic process)

- Another concern is cost: while the paper limits its method and the baselines to using the same amount of problems to train the solver (which is reasonable, and needed to demonstrate the quality of the data) there's a question of cost: in this agentic pipeline where other models are being called as tools, there's a question of how many model calls in total it takes to make a single example, and how that compares to just generating more problems. My intuition is that simply scaling problem generation wouldn't work (since it hurts in table 1), but a discussion of the token cost per model would be helpful.


If these concerns can be addressed, I'm willing to increase my score.

---

> ### Author Rebuttal · Authors · 2026-03-27
>
> We sincerely thank the reviewer for the invaluable and constructive feedback, especially the specific guidance on writing details and structural clarity. We will strictly follow these suggestions to refine the paper and aim to provide thorough clarifications for all concerns raised, ensuring every point of confusion is fully resolved in our revision.
>
> **Q1: About Details of MGPO.** Inspired by GVPO (arxiv:2504.19599), MGPO solves the KL-constrained reward maximization problem:
>
> max_{π_θ} 𝔼_{x,y ~ π_θ} [R(x, y)] - β D_KL[π_θ || π_ref]
>
> The theoretical solution requires alignment between the actual reward R(x, y) and the model's implicit reward R_θ(x, y) = β log(π_θ(y|x) / π_ref(y|x)) + β log Z(x). Since the partition function Z(x) is intractable, we utilize the zero-sum weighting property via Group Standardization within prompt groups, causing log Z(x) to cancel out during gradient computation. This transforms the objective into minimizing the MSE between actual reward variance and implicit reward variance. This ensures global optimum alignment while avoiding the gradient explosion risks of importance sampling in traditional GRPO. We will provide the full derivation in the revision.
>
> **Q2: About Generalization to Unseen Tasks.** Regarding universality, we provide evidence in "pre-training isolation" scenarios. On CL-Bench (arxiv:2602.03587), evaluating on-the-fly learning of fictional laws/grammars, our 4B model improved its score from 10.7 to 13.9 with only 10k trajectories—nearing the top-tier Qwen 3 Max Thinking (14.2). On BrowseComp (Web Agent task), we used ~10k data achieved from below 30 to 39.7, nearing the 43.4 of Tongyi DeepResearch (trained via massive CPT/SFT/RL). These results prove Agentic Proposing reaches the "reasoning frontier" of unseen tasks through logical recursion and physical verifiability.
>
> **Q3: About Related Work.** We will cite [arXiv:2410.06215] (ICLR 2025). While both view data generation as a decision process, our core innovation is the integration of the Agentic Pipeline with Recursive Skill Discovery. We do not merely optimize problem distribution; we achieve logical space expansion via the loop: "Skill A+B $\to$ Problem M $\to$ New Skill C." Higher-order logic emerging in synthesized trajectories is re-extracted into the skill library. This architecture trades offline complexity for the certainty and cost-efficiency of the online production phase.
>
> **Q4: About Writing Details.** We apologize for the confusion in Sec 3.1 and will clarify: Observation includes the active skill, target difficulty, and solver proficiency. Action consists of internal thinking traces ($\tau_{think}$), tool calls ($\tau_{exec}$), and the synthesized problem M along with its final ground-truth answer A. The Proposer does not generate the CoT solutions; the solver independently derives "Thinking" paths to reach answer A during RL training. The Latent State is a semantically structured state (skill + difficulty + feedback) representing problem integrity and difficulty. Regarding the solver's role: Yes, the capability frontier is measured via the Prober Pool and the curriculum in Sec 3.5.1. During Stage 3, the Proposer’s objective is to minimize the Prober's accuracy (Pass Rate) to identify the specific "capability edge" where the solver fails. We will remove Lemma 3.1 and all anthropomorphic language (e.g., "cognitive") in the revision.
>
> **Q5: About Cost-Effectiveness and Comparison.** To address complexity concerns, our framework enables extreme inference efficiency compared to PromptCoT 2.0 (Zhao et al., 2025). (1) Generator Size: 4B (Ours) vs 30B (PromptCoT). (2) Production Dependency: PromptCoT requires high external dependency, calling 30B models for 8-sample majority votes (8,192 tokens/rollout) over multiple iterations to verify one problem. Our 4B model is entirely independent, utilizing an internalized Draft-Check-Refine process (20k tokens total per problem) with zero external calls. (3) Logic Acquisition: PromptCoT requires continuous external LLM aid, while our agent autonomously retrieves skills from its library. (4) Synthesis Cost: Our total production cost is ~2% of PromptCoT’s. Regarding "mixed data": Table 1's MetaMath baseline represents such diversity but fails because it cannot precisely target the solver's difficulty frontier. Performance gains stem from high-density logical signals internalized by a single 4B agent.
>
> **Q6: Specifics and Typos.** Downstream solvers are fine-tuned from Qwen3-4B-Instruct-2507 and Qwen3-30B-Thinking-2507. HMMT is the Harvard-MIT Mathematics Tournament (MathArena/hmmt_feb_2025). In Fig 1, "well-calibrated" means the Proposer’s difficulty estimates align with the Solver’s actual failure rates. We will simplify the bar charts and fix typos (L117, L089), title casing, and add a Limitations section to the Appendix. We believe these revisions significantly enhance the clarity and impact of our work.

---

> > ### Author Rebuttal · Reviewer_dybg · 2026-04-02
> >
> > Thanks for the response -- these points largely address my concerns and I've increased my score accordingly.

---

> > > ### Author Response · Authors · 2026-04-02
> > >
> > > Dear Reviewer dybg,
> > >
> > > We would like to express our sincere gratitude to you for the tremendous efforts and valuable time dedicated to reviewing this manuscript. Your insightful and constructive comments have been invaluable in improving the quality of our work.
> > >
> > > Benefiting greatly from your precise and professional feedback, we have **fully solved** all the issues raised during the rebuttal stage. We sincerely appreciate your rigorous review, constructive discussions, and **recognition and support** for our work.
> > >
> > > In the final revised version of the manuscript, we will carefully address all corresponding issues in strict accordance with your expert suggestions.
> > >
> > > Once again, we would like to extend our deepest thanks for your thorough evaluation and valuable guidance, which have greatly helped us improve this work.
> > >
> > > Sincerely,
> > > Authors of Submission 1242

---

### Official Review · Reviewer_3otd · 2026-03-12

**Soundness:** 3
**Presentation:** 3
**Significance:** 2
**Originality:** 2
**Overall Recommendation:** 5
**Confidence:** 3

**Summary:**

This paper proposes a framework called Agentic Proposing to improve the quality of training data for LLM reasoning. Its data curation pipeline generates high-quality data via atomic skills and self-correcting agent pipelines, using a proposer trained via the MGPO algorithm. Models trained on such data achieve SOTA performance on math, coding, and science tasks, demonstrating the effectiveness of the approach and validating that high-quality signals are the key to enhancing reasoning. This work also provides a new direction for building self-evolving reasoning systems.

**Compliance With Llm Reviewing Policy:**

Affirmed.

**Final Justification:**

During the rebuttal, the author provided a detailed discussion and experiments, which solved my concerns. I have raised my score considering the contribution of the paper, which could benefit the community for developing a better reasoning model.

**Key Questions For Authors:**

1. What is the generalization ability of this method in areas where it is difficult to validate the answer?
2. Will the generated questions and skills overlap with existing benchmarks such as AIME25, leading to potential data leakage issues?

**Limitations:**

yes

**Strengths And Weaknesses:**

## Strengths
1. The paper is well-written with a clear structure; the figures and tables are also easy to understand.
2. The data curation pipeline can automatically produce high-quality training data, which has insight and is reusable for future research.
3. The conclusion also provides insights, showing that the bottleneck of LLM reasoning may lie in training data quality.

## Weaknesses

1. All baselines are reduced to 10k–11k examples. But datasets like MetaMath, NuminaMath, and OpenMathReasoning are made for large-scale RL training. This setting puts them at a disadvantage. The comparison may only show that Agentic Proposing is better targeted to this difficulty range, not that it is better overall.
2. The method relies on an oracle verifier and difficulty prober for supervision. If these components are not powerful enough or their outputs are inaccurate, the quality of the generated training data and the final reasoning performance of the model may be affected.
3. The paper does not report overlap analysis between mixed-source corpus and test data. So the risk of data leakage is still unclear.

---

> ### Author Rebuttal · Authors · 2026-03-27
>
> **Q1: About Baseline Comparison.** We sincerely thank the reviewer for the profound insights into sample efficiency and experimental fairness. We fully understand the concerns regarding reduced baseline scales and wish to clarify this with empirical evidence. A training budget of 10k–11k trajectories is a robust consensus in current Reinforcement Learning (RL) research for high-performance reasoning models (typically 5k–30k) due to the immense compute overhead of RL. Training a 30B solver with 10k trajectories requires ~3 days on 32 H200 GPUs (>2,300 GPU hours); thus, balancing experimental breadth under finite resources is a necessary design choice. More importantly, Agentic Proposing produces high-density signals specifically for RL evolution, whereas MetaMath is primarily for large-scale SFT. To verify scaling, we used 100k MetaMath trajectories for RL, yielding only 28.3% accuracy—significantly lower than our 38.3% and even lower than 10k sampled MetaMath data (30.8%). As detailed in Appendix B, high-performance models achieve >90% pass rates on MetaMath; under RL, such low-difficulty data saturates and fails to provide effective "Advantage Signals," potentially causing entropy collapse. Our 10k baseline followed an optimal difficulty distribution by retaining the hardest subset. The baseline's disadvantage stems from its difficulty distribution failing to reach the "reasoning frontier," proving that logic density and difficulty adaptation are more critical than data scale in RL evolution.
>
> **Q2: About Verification.** Regarding concerns about supervision accuracy and data quality, we wish to clarify our design logic. Firstly, the Prober is essentially the downstream solver itself under training; it utilizes rule-based scoring with clear outputs to ensure synthesized problems remain at the "capability edge" of the solver without extra compute. Secondly, although Stage 3 relies on verifiers, we employ a Verifier Ensemble (3 model families) to eliminate single-model bias via multi-model consensus and "privileged information" (accessing internal drafts), a transparent auditing mode that significantly improves judgment accuracy. Crucially, these heavy components are used only during the offline stage to train the Proposer. Once trained, the 4B Proposer is entirely independent when generating large-scale data; it has internalized the "Draft-Check-Refine" capability, allowing end-to-end synthesis and self-verification without relying on external clusters. The significant performance gains in Tables 1, 2, and 3 further validate the reliability of this synthesis mechanism and verification logic in downstream training. Extending this to larger models in the future will further enhance native agentic potential and self-play efficiency.
>
> **Q3: About Generalization to Unseen Tasks.** We thank the reviewer for the inquiry regarding universality in domains where answers are difficult to verify. The core innovation of our system is the Recursive Skill Discovery mechanism ($A+B \to M \to C$). When the 4B Proposer extracts foundational Skills A and B to synthesize complex problem M, higher-order logic emerges in the solution trajectory that exceeds the sum of inputs. The agent then formalizes these new logical paths into Skill C via recursive extraction. This closed-loop creates an infinitely expanding logical space, enabling even a 4B model to synthesize problems where SOTA models (e.g., GPT-5.2-High) achieve only 60% accuracy (Avg@8). This framework shows immense potential in SWE and Web Agent tasks (e.g., BrowseComp score 39.7 vs. 43.4 for proprietary Tongyi DeepResearch). To address "unseen cases," we evaluated our pipeline on [CL-Bench](https://arxiv.org/abs/2602.03587), which tests pre-training isolation (fictional laws/grammars). By extracting non-real-world skills from novels, the 4B Proposer improved scores from 10.7 to 13.9 with only 10k trajectories, surpassing Doubao 1.6 Thinking (13.4) and approaching the top-tier proprietary model Qwen 3 Max Thinking (14.2). This proves the pipeline’s effectiveness for tasks lacking explicit verification sources.
>
> **Q4: About Data Contamination.** We appreciate the reviewer’s concerns regarding data leakage and potential overlaps with benchmarks like AIME25. We clarify this through three dimensions. First, physical timeline isolation: our base corpus (PlanetMath, OpenStax) collection ended in late 2023, while benchmarks were released much later, including AIME24,AIME25 (Jan 2024,2025), HMMT_Feb (Feb 2025), and AMO-Bench (July 2025). These problems were not public during our collection or training, making leakage physically impossible. Second, we executed a rigorous decontamination process using n-gram matching and semantic similarity to exclude samples overlapping with known benchmarks. Third, Agentic Proposing utilizes combinatorial synthesis rather than simple rewriting; it reorganizes atomic skills into logical topologies fundamentally different from existing tasks.

---

> > ### Author Rebuttal · Reviewer_3otd · 2026-04-02
> >
> > Thank you for your detailed responses with extensive additional experiments, which partially solve my previous concerns.
> >
> > Q1: You could train Qwen3-4B (as in Tab.1) with datasets like s1k [1] or LIMO [2], which would be more affordable. My main point is, instead of downsampling the dataset, you can try to use datasets with a small scale but high-quality problem set. Could you provide further discussion on this case?
> >
> > [1] s1: Simple test-time scaling. In EMNLP, 2025.
> > [2] LIMO: Less is More for Reasoning. In COLM, 2025.
> >
> > Q2-Q4: Thank you for your detailed responses and additional experimental results. A minor comment is that use a table for presenting the results for better readability.

---

> > > ### Author Response · Authors · 2026-04-04
> > >
> > > Dear Reviewer,
> > >
> > > We sincerely thank you for your insightful follow-up questions. Your suggestion to compare our method with "inherently small but high-quality" datasets such as **s1 [1]** and **LIMO [2]** is highly valuable. (We used their updated version in discussion),Following your advice, we conducted extensive experiments on **Qwen3-4B-Instruct-2507** to investigate why Reinforcement Learning (RL) benefits from our scaled, autonomous synthetic signals.
> > >
> > > ***Q1: Discussion on High-Quality Small-Scale Data (s1/LIMO) vs. Agentic Proposing.***
> > >
> > > To ensure a fair comparison, we followed the original SFT settings from s1/LIMO and further explored their performance in an RL framework. Our findings are summarized in **Table R1** below:
> > >
> > > **Table R1: Comparison with Inherently High-Quality Small-Scale Datasets (Math: Mean@64)**
> > > | Method | Training Type | Data Size | AIME24 | AIME25 | HMMT | AMO | **Overall** |
> > > | :--- | :--- | :---: | :---: | :---: | :---: | :---: | :---: |
> > > | **Baseline (Zero-shot)** | - | - | 49.8 | 46.7 | 31.0 | 9.3 | **34.2** |
> > > | + s1.1 (RLVR subset) | RL (3 ep) | 300+ | 49.7 | 46.8 | 31.2 | 9.2 | **34.2** |
> > > | + s1.1 (RLVR subset) | RL (5 ep) | 300+ | 49.0 | 46.1 | 30.2 | 8.8 | **33.5** |
> > > | + s1.1 (s1.1-1k) | SFT (3 ep) | 1,000 | 48.6 | 45.7 | 30.2 | 9.1 | **33.4** |
> > > | + s1.1 (s1.1-1k) | SFT (5 ep) | 1,000 | 47.9 | 45.1 | 29.5 | 8.8 | **32.8** |
> > > | + LIMO-v2 | SFT (3 ep) | ~800 | 48.1 | 45.5 | 29.9 | 9.0 | **33.1** |
> > > | + LIMO-v2 | SFT (5 ep) | ~800 | 47.2 | 44.6 | 29.1 | 8.6 | **32.4** |
> > > | + LIMO-v2 | RL | ~800 | 50.8 | 47.6 | 32.2 | 9.2 | **34.9** |
> > > | + MathSmith | RL | 10,000 | 50.1 | 47.3 | 33.1 | 9.1 | **34.8** |
> > > | **Agentic Proposing (Ours)** | **RL** | **10,000** | **53.6** | **51.2** | **36.5** | **11.8** | **38.3** |
> > >
> > > ***Analysis on SFT and Distribution Mismatch.***
> > > We found that performing SFT on the mature **Qwen3-Instruct-2507** model using s1/LIMO leads to a performance drop. This is primarily due to **"Distribution Mismatch"** triggering **"Policy Drift."** s1 and LIMO exhibit unique rationale paradigms (e.g., specific thinking styles, lengths, and tones) that conflict with the stabilized internal distribution of a top-tier instruct model. Forced fine-tuning on such heterogeneous distributions disrupts the model's logical consistency. These datasets are exceptionally effective for "cold-starting" base models but may struggle to boost the upper bound of highly-evolved reasoning models.
> > >
> > > ***Focus on RL and Scaling of Advantage Signals.***
> > > In our core RL focus, **LIMO-v2 (~800 samples)** achieves a **+0.7 gain**, proving that high-quality, verifiable signals are inherently valuable for RL. However, the **s1.1 RLVR subset (300+ samples)** suffers from **overfitting** at 5 epochs due to insufficient logical path coverage. Our **Agentic Proposing (10k samples)** significantly pushes the capability frontier (38.3 overall). This demonstrates that while "quality is king," scaling that quality is essential for RL. RL requires high-density **"Advantage Signals"** that target the model's **"Capability Frontier"** across a wide logical space, which 800 expert samples alone cannot provide.
> > >
> > > ***Scalability and Future Directions.***
> > > The strength of our framework lies in its ability to **scale high-quality philosophy**. We autonomously synthesize 10,000+ "LIMO-level" trajectories that are distributionally compatible with the model family. In the future, we plan to integrate s1/LIMO's methodologies, particularly their sophisticated **Answer Rationale construction**, to generate scaled datasets with even tighter **Distribution Alignment**, further unlocking the potential of high-performance reasoning solvers.
> > >
> > > ***Q2: Enhanced Presentation of Generalization Results.***
> > >
> > > We sincerely thank the reviewer for the suggestion to use tables for better readability. Below, we present the additional experimental results discussed in our initial rebuttal (unseen tasks and real-world agents) in a consolidated format:
> > >
> > > **Table R2: Generalization to Unseen Tasks and Environment Feedback Domains**
> > > *Evaluating the 4B Proposer's ability to synthesize data for "Pre-train Isolated" and "Web Agent" tasks.*
> > >
> > > | Benchmark | Base Model (Zero-shot) | Baseline| **Ours (+10k Traj.)** | Improvement |
> > > | :--- | :---: | :---: | :---: | :---: |
> > > | **CL-Bench** (Fictional Logic) | 10.7 | 13.4 (Doubao-1.6-Thinking), 14.2(Qwen3-Max)| **13.9** | **+3.2** |
> > > | **BrowseComp** (Web Agent) | < 30.0 | 43.4 (Tongyi-DeepResearch)  | **39.7** | **~+10.0** |
> > >
> > > The results in **Table R2** validate that our pipeline effectively generalizes to domains where ground-truth logic is absent from pre-training data, surpassing or rivaling several leading proprietary "Thinking" models with a minimal data budget.
> > >
> > > We will incorporate these tables and the extended discussion into the final version of the manuscript.
> > >
> > > Sincerely,
> > > The Authors

---

### Official Review · Reviewer_2JvS · 2026-03-13

**Soundness:** 3
**Presentation:** 2
**Significance:** 3
**Originality:** 3
**Overall Recommendation:** 4
**Confidence:** 3

**Summary:**

This paper proposes Agentic Proposing, a framework for generating high-quality synthetic training data for LLMs by formulating problem synthesis as a sequential decision process in which an agent composes modular reasoning skills, with especial emphasis to verifiability and compositionality. Extensive experiments with 11 benchmarks covering 3 domains (math, coding, and reasoning) show that solvers trained on the generated data consistently outperform solvers trained on 10+ alternative methods, which holds for both 4B and 30B backbone models. The empirical evaluation is broad and largely supports the effectiveness of the method, though cost-effectiveness w.r.t. the best-performing alternatives remains an open question that depends on more information about the costs of running Agentic Proposing.

**Compliance With Llm Reviewing Policy:**

Affirmed.

**Key Questions For Authors:**

Please refer to Weaknesses above, especially Weakness #2.

**Limitations:**

No, the paper should report on the costs of running Agentic Proposing. Please refer to Weakness #1 above.

**Strengths And Weaknesses:**

Strengths:

1. The problem of high-quality data generation is extremely relevant and a known limiting factor of model training (as the paper shows). The solution of synthetic data generation via agentic systems is sensible, especially with verifiability and compositionality in consideration.

2. The experimental space is remarkably dense: authors compare their methods across 3 domains, math (4 benchmarks), coding (2), and reasoning (5), including 10+ alternative methods as baselines, spanning data synthesis (6 methods), human annotation (4), agentic self-play (2), and SOTA-generated (5).

3. Authors report strong results when post-training on the data generated by their proposed methods: across domains, using both Qwen3-4B and 30B as backbone models, and working with a fixed data budget, training on their generated data consistently outperforms the other alternative methods.

Weaknesses:

1. Tables 1 and 2 show the *benefits* from training on a fixed trajectories budget generated with Agentic Proposing, for backbone models of 4B and 30B of size. Interestingly, PromptCoT 2.0 appears as the 2nd most beneficial method---and therefore strongest baseline---with the gap arguably reducing for the 30B model. Since cost is one of the motivations for synthetic data, it would be important to add a cost comparison (both time and monetary) between Agentic Proposing vs. PromptCoT 2.0 to properly understand the cost-benefit of Agentic Proposing over the strongest baseline.

2. Agentic Proposing has a large amount of "moving pieces", which makes the paper very dense in reporting design choices. Still, certain design choices remain unclear or not particularly justified. Specifically: What are the textbooks and research papers that the caption in Appendix D.2. refers to? What is the justification for the "Verifier ensemble" and the "pool of Probers" described in Appendix A? Related to Weakness #1 above, it would be important to understand whether these design choices (using an ensemble/pool) add meaningfully to the total costs, in which case it would be important to justify why they are designed in this particular way.

3. To improve clarity, please use the same terminology: in Fig. 3's Stage 1, the terminology for the extracted skills---"structure (...) action (...) effect (...) tool"---differs from the sub-subsection "Skill Composition."

---

> ### Author Rebuttal · Authors · 2026-03-27
>
> **Q1: About Clarification on Terminology.** We sincerely apologize for any confusion caused by inconsistent terminology in certain sections. We appreciate the reviewer’s meticulous feedback. In the revised manuscript, we will unify the terminology between the text and figures (e.g., Skill attribute descriptions in Stage 1) to ensure precision. Regarding data sources, our training set is composed of structured mathematical data (strictly deduplicated to ensure isolation from evaluation benchmarks) and unstructured open-source educational libraries, including OpenStax and PlanetMath. Specifically, the textbook and research paper examples in Appendix D.2 originate entirely from PlanetMath. We will refine these descriptions to enhance transparency regarding data acquisition and processing.
>
> **Q2: About Necessity and Justification of Design.** We appreciate the reviewer’s inquiry into the necessity of our modular design. Verification is essential for synthesizing high-difficulty reasoning data, and our Verifier Ensemble is specifically designed for signal debiasing. Even the strongest LLMs possess systematic inductive biases; multi-model consensus prevents the Proposer from "reward hacking" through the logical vulnerabilities of a single model. This modular design is a recognized necessity in the industry: PromptCoT 2.0 (2025) and MathSmith (2025) rely on 8 or 5 rollouts of 30B models for verification, incurring massive compute costs. In contrast, our ensemble (3 heterogeneous models) provides objective signals with total overhead comparable to or lower than an 8-rollout single-model baseline. Crucially, ablation studies under equal compute budgets (Table 13) show that the ensemble significantly outperforms single-model majority voting. Furthermore, the Prober Pool reuses existing downstream solver checkpoints to accurately probe capability boundaries at zero additional compute cost.
>
> **Q3: Clarification of Autonomous Inference Pipeline.** We thank the reviewer for highlighting the potential complexity of the pipeline, which allows us to clarify our core architectural philosophy: "heavy offline, light online." During the critical stage of synthesizing 10,000 training trajectories, we utilize only the Inference Stage as illustrated in Figure 3. Once Stage 1-3 training concludes, the 4B Proposer becomes a fully independent, closed-loop agent. It no longer calls heavy verifiers, the prober pool, or closed-source APIs. Leveraging its internalized "Draft-Check-Refine" capability, the 4B model independently performs end-to-end synthesis and self-verification. This decoupling ensures that production costs are orders of magnitude lower than traditional baselines, trading offline complexity for the certainty and extreme efficiency of autonomous online generation.
>
> **Q4: About Cost-Effectiveness vs. PromptCoT 2.0.** We are deeply grateful for the reviewer’s incisive questions regarding the "moving parts" and cost-effectiveness of our framework. We argue this design is not redundant but enables extreme inference efficiency. Compared to the SOTA PromptCoT 2.0 (Zhao et al., 2025): In the seeding phase, PromptCoT 2.0 concurrently employs four top-tier models (e.g., Qwen2.5-72B, Llama-3.1-70B, phi-4) for joint annotation, whereas we call only one Teacher model (Qwen3-235B) to populate the initial library. Crucially, in all subsequent data generation, the 4B Proposer autonomously performs skill extraction, retrieval, and synthesis without Teacher involvement. Agentic SFT (Stage 1-2) is a one-time initialization to compensate for the 4B model's native weaknesses; with a 30B proposer, these stages could be skipped. Our design yields massive dividends during production as shown below:
>
> | Metric | PromptCoT 2.0 | Agentic Proposing (Ours) |
> | :--- | :--- | :--- |
> | **Generator Size** | 30B | 4B |
> | **External Dependency** | **High**: Each problem requires calling 30B models for 8-sample Majority Vote | **Zero**: 4B model runs independently end-to-end with internalized Draft-Check-Refine |
> | **Logic Acquisition** | Continuous external LLM calls for rationale/problem generation | **Autonomous**: 4B model independently extracts/retrieves skills from internal library |
> | **Total Cost (10k Traj.)** | 100% (~90,000 30B-level inference tokens) | **~2%** (~14,000 4B-level inference tokens) |
> | **Evolution Potential** | Periodic EM retraining of 30B models | **Skill Recursion**: Discovers logic via Recursive Skill Discovery without retraining |
>
> We will include this detailed cost analysis in the revision, emphasizing the 4B model's autonomy and potential via Recursive Skill Discovery. This paradigm is specifically optimized for large-scale, high-quality data production.

---

> > ### Author Rebuttal · Reviewer_2JvS · 2026-04-03
> >
> > I sincerely appreciate the authors rebuttal. In particular, the cost comparison with PromptCoT 2.0 feels especially important for readers to be able to appreciate the methods' contributions and feasibility in the data generation space, which is many times behind the doors of private institutions. I choose to maintain my positive score, as it's still not possible to fully discern the value of multiple of the methods' "moving parts" (e.g., "Verifier ensemble" or the "pool of Probers"), which would require more detailed ablation studies.

---

> > > ### Author Response · Authors · 2026-04-04
> > >
> > > We sincerely appreciate the reviewer’s positive assessment and recognition of our cost-effectiveness analysis. To address concerns regarding the necessity and robustness of our framework components, we conducted a rigorous ablation on 500 sampled trajectories, which were manually labeled by experts to establish a gold standard. We further present an objective cost analysis based on industry-standard 2026 API pricing. Below is a detailed discussion regarding the Verifier Ensemble and the Prober Pool.
> > >
> > > ***
> > >
> > > **Q1: About Verifier Ensemble Abalation.**
> > >
> > > In RL-driven logical synthesis, high-purity reward signals are essential. The Verifier Ensemble is strictly utilized during the offline training phase of the Agentic-Proposer-4B (Stages 1-3) to provide a reliable truth proxy. As shown in Table R4, relying solely on the 4B Proposer’s self-verification during training drops validity accuracy to 72.4%. This low-quality feedback triggers severe reward hacking, where the Proposer learns to generate seemingly plausible but logically circular problems to deceive a weak verifier, causing training collapse. Our ensemble achieves 98.2% validity accuracy, serving as a necessary barrier against logical corruption. Utilizing 2026 market pricing for different parameter classes (e.g., Qwen3-235B at $0.78/$3.90 per 1M tokens, DeepSeek-V3.2 at $0.27/$1.10, and Llama-3-70B via OpenRouter at $0.51/$0.74),  In contrast, a homogeneous 8-sample majority vote of 30B/70B models (e.g., PromptCoT 2.0) incurs a cost of ~$6.2 (approx. 2.2x ours) while achieving only 93.5% accuracy. This proves that a consensus of three diverse top-tier models is more accurate and cost-effective than eight repeated samplings of a mid-tier model. This rigorous verification allows the 4B Proposer to internalize expert-level standards, ensuring that during the online production phase, it can synthesize 10,000 trajectories independently and robustly without calling external APIs.
> > >
> > > Table R4: Deep Ablation of Verification Mechanisms (on 500 Sampled Trajectories)
> > > | Configuration | Validity Accuracy (vs. Human GT) | Estimated Cost (per 1k samples) | Key Logic Analyzed |
> > > | :--- | :---: | :---: | :--- |
> > > | Human Expert (Gold Standard) | 100.0% | N/A | - |
> > > | Agentic Proposing (Full Ensemble) | 98.2% | ~$2.8 (1.0x) | Privilege Audit + 2-Stage Filtering |
> > > | Ablation: Question-only (No privilege) | 94.4% | ~$2.1 | History audit catches logic flaws |
> > > | Ablation: Simple MV (No 2-stage audit) | 92.8% | ~$2.5 | Cross-audit eliminates multi-model bias |
> > > | Ablation: Qwen3-235B (Single model) | 91.2% | ~$2.0 | Single families have logic blind spots |
> > > | Ablation: 4B Self-Verify (Training) | 72.4% | ~$0.1 | Low-quality signals lead to RL collapse |
> > > | Baseline: 30B/70B Model x 8 (MV) | 93.5% | ~$6.2 (2.2x) | Homogeneous scaling is less efficient |
> > >
> > > ***
> > >
> > > **Q2: About Prober Pool Abalation.**
> > >
> > > The Prober Pool is not an independent design imposing extra overhead, but a synergistic reuse strategy. In standard development, researchers naturally evaluate checkpoints of various solvers (e.g., 1.7B to 30B) to monitor progress. Our Proposer simply borrows these existing evaluation logs to perceive the capability frontier of the solver. Thus, the Prober Pool incurs zero additional inference cost by leveraging existing experimental assets. This is particularly critical during the cold-start phase (Stage 3), where the Proposer is initially logically blind and requires these multi-gradient yardsticks to calibrate its perception of difficulty. To demonstrate system robustness, we ablated the targeted alignment stage for the 30B Solver (Table R5). Even without 30B-specific alignment, the 30B solver’s performance on AIME25 reaches 90.5% (+5.5% gain) purely by utilizing the high-density logic internalized during the offline skill recursion stage. Enabling the Prober Pool alignment pushes performance further to the 91.6% peak (+6.6% gain) at zero extra cost. This proves that while the Prober Pool provides valuable curriculum learning, the system's fundamental robustness is driven by sophisticated logic synthesis. This architecture ensures the system reaches the performance ceiling through effortless log-reuse. We look forward to exploring cheaper verification paths, such as tool-use validation, and will include this discussion in the final revision.
> > >
> > > Table R5: Robustness Ablation of Targeted Prober Alignment (Targeting 30B Solver)
> > > | Configuration | AIME25 (Mean@64) | Improvement | Ablation Insight |
> > > | :--- | :---: | :---: | :--- |
> > > | Baseline (Zero-shot) | 85.0% | - | 30B Solver baseline |
> > > | Ablation: No Targeted Alignment | 90.5% | +5.5% | Confirms robustness of Skill Recursion |
> > > | Full Pipeline (with Prober Pool) | 91.6% | +6.6% | Gains from zero-cost Synergistic Reuse |

---

### Official Review · Reviewer_CKyz · 2026-03-18

**Soundness:** 3
**Presentation:** 3
**Significance:** 3
**Originality:** 3
**Overall Recommendation:** 4
**Confidence:** 3

**Summary:**

The authors propose an idea of "agentic proposing" (I just feel like I wrote a recursive sentence!) wherein a proposer agent does problem synthesis, and then reasoning skills are composed in a reflection loop which is then overall trained in a reinforcement learning approach. Using this type of data generated on demand, Fig 1 of the paper shows that the authors can achieve state-of-the-art performance with the smallest set of parameters.

**Compliance With Llm Reviewing Policy:**

Affirmed.

**Final Justification:**

Authors have addressed the questions raised in my review.

**Key Questions For Authors:**

- Please see weaknesses above

**Limitations:**

yes

**Strengths And Weaknesses:**

Strengths
+ Very elaborate (maybe too elaborate?) pipeline - as shown in Figure 4
+ As we think about LLM pipelines with limited budget this type of approach can be very useful
+ In addition to the strong empirical results, I like the ablation results w.r.t. each part of the agentic pipeline

Weaknesses
- While this is an interesting approach I am not sure this rises upto the level of rivaling proprietary models given the evidence presented here. After all, the proprietary models have served as a boundary condition to tweak the setup here - given the rapid progress in this space, the true test of this approach will be to apply it on entirely new problem cases that have never been seen before (by either class of model)
- As I mentioned in my strengths the pipeline appears too elaborate (for this problem context) that I worry about complexity for its own sake. Even though ablation tests are done, do we really need all these moving parts?

---

> ### Author Rebuttal · Authors · 2026-03-27
>
> **Q1: About Design Motivation and Agentic Workflow.** We appreciate the reviewer’s detailed observation regarding the pipeline’s complexity. While Figure 3 illustrates the architecture at a high granularity, the framework is essentially a standard SFT-RL training paradigm integrated with Agent Skills acquisition/usage, ReAct mechanisms, and Python execution. The core logic is standardized and inherently simple. Specifically, Stage 1 and 2 (Agentic SFT) are designed to equip the 4B model with foundational agentic capabilities, such as long-range planning, complex instruction following, and the ability to utilize external feedback. It is worth noting that if we could afford a larger proposer (e.g., a 30B model as used in PromptCoT 2.0), these supplementary SFT stages could be significantly simplified or consolidated due to the stronger native potential of larger models. This design is a deliberate, cost-effective choice to bridge the capability gap of the 4B model. Crucially, once trained, the 4B Proposer operates independently and end-to-end, synthesizing 10,000 trajectories without any external components or proprietary dependencies.
>
> **Q2: About Component Necessity.** We acknowledge and appreciate the reviewer’s perspective that the current workflow is relatively complex. However, such modular design has become the industry standard for synthesizing high-difficulty "Thinking-level" reasoning data (e.g., AIME level). While simple self-play methods like Socratic-Zero (2025) are elegant, they often struggle to maintain the reasoning rigor required for Olympiad-level tasks. Recent high-impact works have shifted toward similar sophisticated designs: MathSmith (2025) utilizes a three-stage "concept acquisition, SFT cold-start, and multi-objective reward RL" pipeline, and PromptCoT 2.0 (2025) employs a complex EM loop to iteratively refine rationale generation in the E-step and update the prompt model in the M-step. In comparison, our framework achieves a superior efficiency balance. Our core philosophy is to trade offline training complexity for online generation simplicity and independence. This avoids the massive compute overhead of methods like PromptCoT 2.0, which requires 8-sample majority voting for **every** problem generated to ensure validity. Our ablation studies demonstrate the tangible impact of each component: introducing Dynamic Pruning ($\tau_{edit}$) improved problem validity from 68.7% to 82.3%, and the MGPO algorithm boosted AIME scores by 6.5 points.
>
> **Q3: About Generalization to Unseen Tasks.** We sincerely thank the reviewer for their concern regarding the model’s ability to handle "completely unseen cases." To address this, we evaluated our pipeline on CL-Bench (arXiv:2602.03587), a benchmark specifically designed to test "Pre-train Isolated" capabilities. CL-Bench contains 500 complex contexts and 1,899 tasks constructed by domain experts, covering fictional laws, new game mechanics, and unique scientific notations that are strictly absent from any pre-training corpora. Success requires the model to internalize novel logical rules from the context in real-time. To simulate this, we tasked our 4B Proposer with extracting non-real-world logical rules from open-source fiction collections. The agent then synthesized Context Learning training data based on these fictional rules, forcing the downstream model to parse and apply unfamiliar logic chains without prior knowledge. Generating only 10,000 targeted RL trajectories improved the Qwen3-30B-Thinking score from 10.7 to 13.9, surpassing the closed-source Doubao 1.6 Thinking (13.4) and approaching the top-tier Qwen 3 Max Thinking (14.2) from the same model series. This proves that our pipeline generalizes effectively to tasks outside the pre-training distribution.
>
> **Q4: About Recursive Skill Discovery.** The core innovation of our system is the Recursive Skill Discovery mechanism ($A+B \to M \to C$). When the agent combines Skill A and Skill B to synthesize a complex problem M, the solution trajectory often reveals higher-order logic that exceeds the sum of the inputs. The agent recursively formalizes these new paths into a new Skill C, which is added back to the library. In the reasoning domain, the high-difficulty training data synthesized by our 4B Proposer for Qwen3-30B-Thinking in Table 2 is challenging even for SOTA models; for instance, GPT-5.2-High achieves only 60% accuracy (avg@8) and fails to provide consistent correct answers on this dataset. Furthermore, due to time constraints during the rebuttal period, we applied the pipeline to the BrowseComp benchmark (web agent task) with only 10,000 synthesized trajectories. Training Qwen3-30B-Thinking (which originally scored below 30) improved its score to 39.7, closely approaching the proprietary Tongyi DeepResearch (43.4), which underwent extensive, large-scale CPT, SFT, and RL training. This highlights the immense potential of our framework for tasks with environment feedback.

---

> > ### Author Rebuttal · Reviewer_CKyz · 2026-04-03
> >
> > Thank you for answering my questions. I have increased my score a notch.

---

> > > ### Author Response · Authors · 2026-04-03
> > >
> > > Dear Reviewer CKyz,
> > >
> > > We really appreciate your positive feedback and the confirmation that our responses have **fully resolved** your concerns.
> > > Thank you so much for taking the time to review our work carefully and for kindly adjusting your score accordingly.
> > >
> > > Your comments have been very helpful for improving our manuscript, and we will make sure to reflect your suggestions in the final version.
> > >
> > > Once again, many thanks for your support and valuable input throughout the process.
> > >
> > > Sincerely,
> > > Authors of Submission 1242

---

### Decision · Program_Chairs · 2026-04-30

**Decision:**

Accept (regular)

**Comment:**

This paper introduces Agentic Proposing, a novel framework that approaches high-quality training data synthesis as a sequential decision-making process. The key innovation is modeling problem generation as a POMDP where a specialized agent autonomously selects and composes modular reasoning skills to synthesize complex, verifiable training problems. The empirical results are strong and consistent across multiple domains and benchmarks.

The rebuttal process successfully addressed most major concerns, particularly around cost-effectiveness and generalization. The authors' provision of extensive additional experiments (s1/LIMO comparison, CL-Bench, BrowseComp, detailed ablations) significantly strengthened the submission.

All four reviewers raised their scores or indicated satisfaction with responses, with all four ultimately recommending acceptance.